# HEATACO: Heatmap-Guided Ant Colony Decoding for Large-Scale Travelling Salesman Problems

## Abstract

Heatmap-based non-autoregressive solvers for large-scale Travelling Salesman Problems output dense edge-probability scores, yet final performance largely hinges on the decoder that must satisfy degree-2 constraints and form a single Hamiltonian tour. Greedy commitment can cascade into irreparable mistakes at large $N$, whereas MCTS-guided local search is accurate but compute-heavy and highly engineered. We instead treat the heatmap as a soft edge prior and cast decoding as probabilistic tour construction under feasibility constraints, where the key is to correct local mis-rankings via inexpensive global coordination. Based on this view, we introduce HEATACO[1], a plug-and-play Max–Min Ant System decoder whose transition policy is softly biased by the heatmap while pheromone updates provide lightweight, instance-specific feedback to resolve global conflicts; optional 2-opt/3-opt post-processing further improves tour quality. On TSP500/1K/10K, using heatmaps produced by four pretrained predictors, HEATACO+2opt achieves gaps down to 0.11%/0.23%/1.15% with seconds-to-minutes CPU decoding for fixed heatmaps, offering a better quality–time trade-off than greedy decoding and published MCTS-based decoders. Finally, we find the gains track heatmap reliability: under distribution shift, miscalibration and confidence collapse bound decoding improvements, suggesting heatmap generalisation is a primary lever for further progress.

---
[1]Anonymous Institution, Anonymous City, Anonymous Region, Anonymous Country. Correspondence to: Anonymous Author <anon.email@domain.com>.

Preliminary work. Under review by the International Conference on Machine Learning (ICML). Do not distribute.

[1]An anonymized repository is available at https://anonymous.4open.science/r/HeatACO.

## 1. Introduction

The Travelling Salesman Problem (TSP) (Reinelt, 1991) remains a central benchmark for combinatorial optimisation (Korte et al., 2011) and learning-based routing (Bengio et al., 2021; Liu et al., 2023). While early neural solvers typically targeted $N \leq 100$ nodes (Kool et al., 2019; Bresson & Laurent, 2021), recent heatmap-based non-autoregressive solvers routinely evaluate at $N \geq 500$ and scale up to $N = 10^4$ nodes (Fu et al., 2021; Qiu et al., 2022; Min et al., 2023; Sun & Yang, 2023; Pan et al., 2025). In this paper, we refer to instances with $N \geq 500$ nodes as *large-scale*. Such scales are increasingly relevant in routing and logistics (Davendra, 2010; Zhou et al., 2025a). While exact solvers and powerful heuristics (e.g., Concorde (Applegate et al., 2006) and LKH-3 (Helsgaun, 2017)) can produce excellent tours, they typically rely on handcrafted search and become costly as $N$ grows (Ilavarasi & Joseph, 2014). In parallel, neural combinatorial optimisation (NCO) (Bengio et al., 2021; Liu et al., 2023) aims to amortise computation by learning reusable inductive biases from data (Vinyals et al., 2015; Kool et al., 2019).

A prominent NCO paradigm is *heatmap-based non-autoregressive* (NAR) solvers (Sun & Yang, 2023; Fu et al., 2021; Qiu et al., 2022; Min et al., 2023). A neural network predicts a dense edge-confidence matrix $H$ in one forward pass, and a downstream decoder converts $H$ into a feasible tour. This decoupling is attractive for large instances because inference is parallel; however, it elevates decoding into the decisive bottleneck (Xia et al., 2024; Pan et al., 2025). The overall prediction–decoding pipeline is summarised in Figure 1. Greedy edge-merging decoders (Li et al., 2023; 2025) are fast but brittle: early local decisions can violate the degree-2 and single-cycle constraints and lead to large gaps (Xia et al., 2024). The current state-of-the-art heatmap decoders combine the heatmap with MCTS-guided $k$-opt local search (Fu et al., 2021; Qiu et al., 2022). These pipelines have been evaluated at problem sizes up to $N = 10^4$ nodes. However, under the fixed per-instance time budgets used for MCTS-guided decoding in prior work, they consume longer computational time than LKH-3 and still leave a noticeable optimality gap. Moreover, their performance depends on carefully engineered action spaces to guide the $k$-opt exchanges (Fu et al., 2021) and extensive hy-

perparameter tuning (Pan et al., 2025), making them costly to adapt across heatmap sources and problem scales.

This paper focuses strictly on the *heatmap-to-tour decoding* stage. Once a heatmap predictor is fixed, the decoder largely determines the achievable quality–time trade-off: greedy merge is fast but brittle, while MCTS-guided $k$-opt can be strong but requires heavy computation and careful action-space engineering. We view the heatmap as a soft prior over edges and decode by constrained sampling under degree and subtour constraints. Since heatmap scores are noisy and often globally inconsistent, this formulation avoids over-committing to spuriously confident edges and enables an explicit exploration–exploitation trade-off with lightweight, instance-specific feedback that iteratively reshapes the edge distribution toward feasible tours. Ant Colony Optimisation (ACO), and in particular the Max–Min Ant System (MMAS) (Stützle & Hoos, 2000), naturally maintains an instance-specific sampling distribution that is refined by solution feedback.

We propose HEATACO, a modular MMAS decoder that injects the heatmap as a multiplicative prior in the transition probabilities while pheromone updates act as lightweight global feedback: edges that repeatedly appear in short tours are reinforced, while spuriously confident edges are progressively suppressed. Because HEATACO operates on sparse candidate lists and admits multi-threaded sampling, it scales to large instances; optional 2-opt/3-opt provides a standard tightening knob. Across heatmaps inferred by multiple pretrained predictors (AttGCN (Fu et al., 2021), DIMES (Qiu et al., 2022), UTSP (Min et al., 2023), DI-FUSCO (Sun & Yang, 2023)), HEATACO+2opt reaches gaps as low as 0.11%/0.23%/1.15% on TSP500/1K/10K and improves the quality–time trade-off relative to greedy and published MCTS-guided decoding (Pan et al., 2025). On TSP500/1K, HEATACO+2opt runs in seconds and, to our knowledge, achieves the best reported gaps among learning-based methods; with 3-opt, we further reduce gaps by 1∼2 orders of magnitude while remaining in the seconds regime.

To clarify when heatmap decoding is effective, we analyse how lightly filtered heatmaps distribute candidate edges across confidence intervals and how edges of the benchmark tours concentrate within these intervals, including out-of-distribution TSPLIB instances (Section 5 and Appendix E). Extended related work is deferred to Appendix H.

Our contributions are:

- We formulate heatmap-to-tour decoding as constrained sampling from a soft edge prior and introduce HEAT-ACO, a modular MMAS decoder that incorporates the heatmap as a multiplicative prior while using pheromone feedback for lightweight, instance-specific global conflict resolution; optional 2-opt/3-opt integrates seamlessly for tour tightening.

- Across four pretrained heatmap predictors and instance sizes up to $N{=}10,000$, HEATACO delivers strong quality–time trade-offs: given fixed heatmaps, HEAT-ACO+2opt reaches gaps as low as 0.11%/0.23%/1.15% on TSP500/1K/10K under seconds-to-minutes CPU decoding budgets, outperforming greedy decoding and improving over published MCTS-guided decoding under comparable or smaller budgets.

- We provide an analysis of applicability and failure modes: HeatACO benefits most when heatmaps preserve a meaningful ranking over a sparse high-recall candidate set, while OOD miscalibration and confidence collapse can limit decoding performance; we support these claims with ablations, convergence, and sensitivity studies, and offer practical guidance for tuning the heatmap strength $\gamma$.

## 2. Background and Problem Setup

### 2.1. Travelling Salesman Problem

We consider TSP instances specified by planar node coordinates (Davendra, 2010; Reinelt, 1991). Given a set of $N$ nodes with coordinates $\{\mathbf{x}_i \in \mathbb{R}^2\}_{i=1}^N$, let $d_{ij} = \|\mathbf{x}_i - \mathbf{x}_j\|_2$ be the pairwise $\ell_2$ distance between nodes $i$ and $j$. Let $V \triangleq \{1, \ldots, N\}$ and $E_{\text{full}} \triangleq \{(i,j) \mid 1 \le i < j \le N\}$ denote the nodes and edges of the complete graph. A tour can be represented by a permutation $\pi$ of $\{1, \ldots, N\}$, with length

$$\mathcal{L}(\pi) = \sum_{t=1}^N d_{\pi_t, \pi_{t+1}}, \quad \text{where } \pi_{N+1} \triangleq \pi_1. \quad (1)$$

The goal is to find the minimum-length Hamiltonian cycle:

$$\pi^\star \in \arg\min_{\pi \in \mathcal{S}_N} \mathcal{L}(\pi), \quad (2)$$

where $\mathcal{S}_N$ is the set of all permutations. TSP is NP-hard; therefore, large instances are typically solved by heuristics (Laporte, 1992; Ilavarasi & Joseph, 2014; Xin et al., 2021).

A tour can also be viewed as an undirected edge set. Let $A \in \{0,1\}^{N \times N}$ be its adjacency indicator. The degree constraint is

$$\sum_{j=1}^N A_{ij} = 2, \quad \forall i \in \{1, \ldots, N\}, \quad (3)$$

and a feasible tour additionally forms a single cycle.

### 2.2. Heatmap-based Non-autoregressive Neural Solvers

Heatmap-based non-autoregressive TSP solvers predict, in one forward pass, a dense pairwise confidence matrix

$$H = f_\theta(\{\mathbf{x}_i\}_{i=1}^N) \in [0,1]^{N \times N}, \quad (4)$$

where $H_{ij}$ is interpreted as the model's confidence that edge $(i,j)$ belongs to a (near-)optimal tour (Joshi et al., 2019; Fu et al., 2021; Qiu et al., 2022; Min et al., 2023; Sun &

Yang, 2023). In practice, $H$ is typically symmetrized (Sun & Yang, 2023), e.g.,

$$H \leftarrow \frac{1}{2}(H + H^{\top}). \tag{5}$$

This modeling choice decouples *prediction* from *construction*: the neural network outputs a soft, dense "heatmap", and a separate decoding algorithm converts $H$ into a feasible tour. Many recent architectures (e.g., GNN (Joshi et al., 2019; Fu et al., 2021) and diffusion-based models (Sun & Yang, 2023; Li et al., 2023; Wang et al., 2025)) fall into this family; our work focuses on this shared decoding bottleneck rather than the specific network $f_\theta$.

### 2.3. The Heatmap-to-Tour Decoding Problem

Given an instance (distance matrix $D$) and a predicted heatmap $H$, the decoder aims to output a feasible tour:

$$\hat{\pi} = g(H, D). \tag{6}$$

The core challenge is that while $H$ provides valuable edge-level signals, feasibility requires a *globally consistent* degree-2 single-cycle structure. As a result, naive edge selection can be severely hindered by conflicts induced by the degree constraint in Eq. (3) and by subtour formation. We analyse how modern heatmap predictors allocate confidence and how edges of the benchmark tours used for evaluation (optimal for TSP500/1K, best-known for TSP10K) concentrate in a narrow confidence band in Section 5.2 (Figure 3), and relate this phenomenon to decoding robustness and distribution shift.

### 2.4. Common Decoding Baselines

**Greedy merge (constructive decoding).** A widely used baseline computes an edge score that combines the heatmap and distances, e.g.,

$$s_{ij} = \frac{H_{ij}}{d_{ij}}, \tag{7}$$

sorts candidate edges by $s_{ij}$, and inserts edges one-by-one subject to feasibility constraints (degree $\leq 2$ and avoiding premature small cycles) (Sun & Yang, 2023). While fast, this approach is brittle on large instances because early local choices can cause global dead-ends under Eq. (3).

**MCTS-guided local search (improvement decoding).** Another line of work constructs an initial tour (often via a greedy method) and then improves it with local search (e.g., $k$-opt) (Fu et al., 2021), where a Monte Carlo Tree Search (MCTS) (Browne et al., 2012) procedure selects promising moves (Xia et al., 2024). Heatmap information is commonly used to bias move proposals toward high-confidence edges. Although effective on moderate problem sizes, the search space and the sequential nature of tree expansion can make the method costly for large $N$, and its performance is sensitive to nontrivial design choices (move generation, rollout policy/value, exploration constants, etc.) (Pan et al., 2025).

**Our focus.** In this paper, we keep the heatmap predictor $f_\theta$ fixed and study the decoding problem: *Given a heatmap that already contains strong structural signals, how can we convert it into a high-quality tour quickly and reliably at large scale?*

## 3. Method

We propose HEATACO, a modular decoder that takes a neural edge-confidence heatmap $H$ and the distance matrix $D$ as input and outputs a feasible TSP tour. HeatACO is built upon the standard MMAS (Stützle & Hoos, 2000) and differs from vanilla MMAS in one targeted way: we inject the heatmap as a multiplicative *prior* in the transition probabilities used during tour construction.

Algorithmically, HeatACO follows the simple pipeline in Figure 1: (i) build sparse candidate lists from the filtered heatmap (optionally augmented with nearest-neighbour edges for connectivity); (ii) run MMAS on these candidates with heatmap-biased transition probabilities; and (iii) optionally apply a fast local improvement step (2-opt/3-opt) and reinforce pheromones with the improved tour. Key implementation details are summarised in Appendix B.

### 3.1. Max–Min Ant System for TSP

MMAS maintains a pheromone matrix $\tau \in \mathbb{R}_+^{N \times N}$ and repeatedly samples tours with a stochastic policy that combines pheromones with a distance heuristic $\eta_{ij} = 1/d_{ij}$. After each iteration, pheromones are evaporated, reinforced on an elite tour, and clamped to $[\tau_{\min}, \tau_{\max}]$ (Stützle & Hoos, 2000). We keep the MMAS update unchanged and inject the heatmap only via the transition rule in Section 3.2.

### 3.2. Heatmap-biased transition probabilities

Given a heatmap $H$ defined in Section 2.2, we treat it as a *soft prior* rather than a calibrated probability and incorporate it into MMAS by biasing the sampling policy. As illustrated in Figure 1, HEATACO uses $H$ to bias sampling toward plausible edges, while pheromone updates provide instance-specific feedback to resolve global inconsistencies. Concretely, we clip and floor the heatmap to suppress the low-confidence tail (using $\epsilon_h = 10^{-4}$ (Fu et al., 2021; Sun & Yang, 2023) and $\varepsilon = 10^{-9}$) and build sparse MMAS candidate lists from high-confidence edges with nearest-neighbour fallbacks to preserve connectivity (Appendix B).

$$\tilde{H}_{ij} \triangleq \begin{cases} H_{ij}, & H_{ij} \geq \epsilon_h, \\ \varepsilon, & \text{otherwise.} \end{cases} \tag{8}$$

Given an ant at node $i$ with unvisited set $\mathcal{U}$, the transition probability to a feasible next node $j \in \mathcal{U}$ is

$$p_{i \to j} = \frac{(\tau_{ij})^{\alpha} (\eta_{ij})^{\beta} \left(\tilde{H}_{ij}\right)^{\gamma}}{\sum_{k \in \mathcal{U}} (\tau_{ik})^{\alpha} (\eta_{ik})^{\beta} \left(\tilde{H}_{ik}\right)^{\gamma}}, \tag{9}$$

where $\eta_{ij} = 1/d_{ij}$ is the distance heuristic and $\alpha, \beta$ are standard MMAS hyperparameters (Stützle & Hoos, 2000). The exponent $\gamma$ controls the strength of heatmap guidance: $\gamma = 0$ recovers vanilla MMAS, while larger $\gamma$ follows the

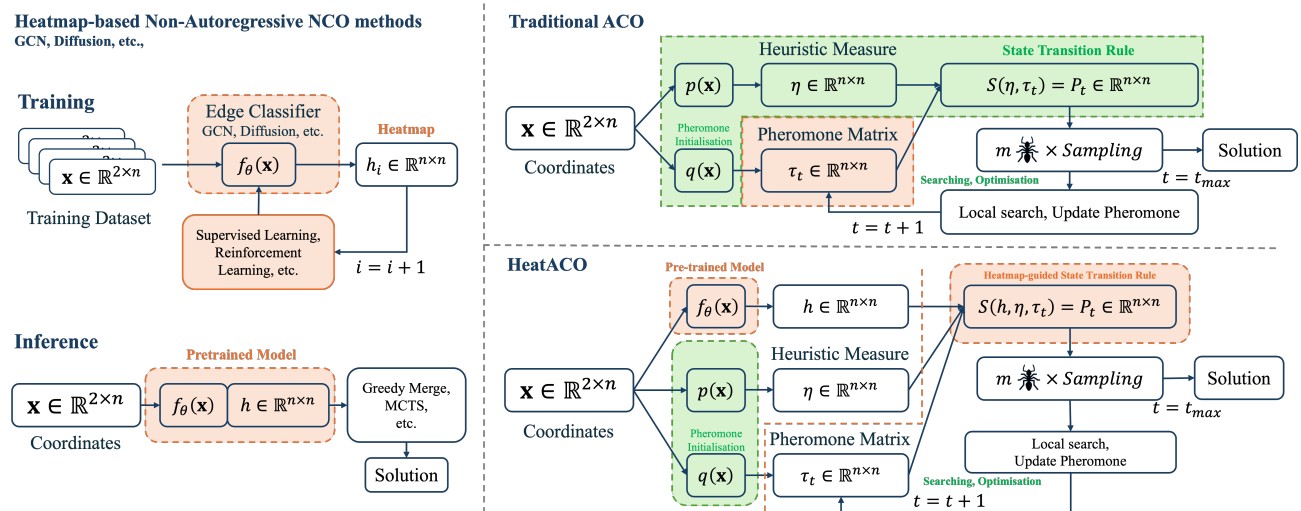

*Figure 1.* Overview of heatmap-based decoding. **Left:** heatmap prediction (NAR), producing an edge-confidence matrix $H$ from node coordinates. **Top-right:** standard ACO decoding constructs a tour using distance heuristics and pheromone feedback. **Bottom-right:** HEATACO injects $H$ as a soft prior into ACO/MMAS sampling while preserving pheromone feedback to correct globally inconsistent local choices.

heatmap ranking more closely. We also provide a label-free heuristic to choose $\gamma$ by targeting the entropy (effective support size) of the heatmap-induced proposal distribution; see Appendix G.2.

### 3.3. Optional local improvement and implementation

HeatACO can optionally apply 2-opt/3-opt local improvement to constructed tours and reinforce pheromones on the improved tours. For scalability, both tour construction and local improvement operate on sparse candidate lists induced by the filtered heatmap with distance-based fallback; implementation details and complexity are summarised in Appendix B.

**Complexity.** Let $N$ be the number of nodes, $m$ the number of ants, $I$ the number of iterations, and $k$ the per-node candidate-list size (default $k=20$ (Stützle & Hoos, 2000)). Preprocessing is dominated by $O(N^2)$ time and memory for distances and heatmap processing. Each MMAS iteration constructs $m$ tours in $O(mNk)$ time; with local improvement, the cost becomes $O(mp_2Nk)$ for 2-opt or $O(mp_3Nk^2)$ for 3-opt (restricted to candidate edges). Pheromone updates are $O(Nk)$ with candidate-based evaporation, and memory is dominated by the distance and pheromone matrices $O(N^2)$. Appendix B.1 provides a code-level breakdown.

## 4. Experiments

### 4.1. Datasets

We use the standard TSP benchmarks released with AttGCN (Fu et al., 2021): **TSP500**, **TSP1K**, and **TSP10K** with $N \in \{500, 1000, 10000\}$. We evaluate on the released instances and report gaps to the benchmark tour length $L^\star$. Additional dataset and benchmark details are in Appendix B.2. For out-of-distribution (OOD) evaluation, we additionally test on three structured TSPLIB instances (pcb442, pr1002,

pr2392); full results and diagnostics are in Appendix E.

We treat the heatmap predictor as fixed and benchmark only the decoder given a precomputed heatmap. We infer heatmaps by running released pretrained predictors on the benchmark instances and evaluate decoding given fixed heatmaps; full provenance is in Appendix B.2.

### 4.2. Compared methods

We compare with classical solvers/heuristics (Concorde (Applegate et al., 2006), Gurobi, LKH-3 (Helsgaun, 2017)) and representative NCO baselines (AM (Kool et al., 2019), LEHD (Luo et al., 2023), DeepACO (Ye et al., 2023), and GLOP (Ye et al., 2024)) from prior work for context. Our primary comparison is among *decoders* for fixed heatmaps: (i) greedy merge (NAR+Greedy) (Sun & Yang, 2023); (ii) the published MCTS-guided $k$-opt decoder (Pan et al., 2025) (a tuned, highly-parallel variant building on AttGCN+MCTS (Fu et al., 2021)); this decoder is carefully tuned with per-heatmap and per-scale hyperparameters, and we follow the released configurations for each heatmap source and problem size; (iii) vanilla MMAS (Stützle & Hoos, 2000) ($\gamma=0$) to isolate the role of the heatmap prior; and (iv) HEATACO (with optional 2-opt/3-opt). Appendix B.2 provides baseline sources and runtime notes.

### 4.3. Metrics

We report three metrics commonly used in prior work: (i) **average tour length** $L$ (lower is better); (ii) **optimality gap** (%) relative to the benchmark tour length $L^\star$, computed as $(L - L^\star)/L^\star \times 100\%$; and (iii) **per-instance runtime** measured as wall-clock time in seconds.

### 4.4. Evaluation protocol

We run MMAS/HEATACO with $m=32$ ants and a fixed budget of $I=5{,}000$ iterations, and report averages over 10

*Table 1.* Main results on the TSP500/1K/10K benchmarks (Fu et al., 2021). We report average tour length $L$, gap (%) to the benchmark tour length $L^\star$, and per-instance wall-clock time $T$; $T$ reporting follows Section 4.4. Baseline provenance and runtime notes are in Appendix B.2. For the numbers/results in the table, lower is better.

| Method | Type | TSP 500 | | | TSP 1K | | | TSP 10K | | |
|---|---|---|---|---|---|---|---|---|---|---|
| | | $L\downarrow$ | Gap(%)$\downarrow$ | $T\downarrow$ | $L\downarrow$ | Gap(%)$\downarrow$ | $T\downarrow$ | $L\downarrow$ | Gap(%)$\downarrow$ | $T\downarrow$ |
| Concorde | Exact | 16.55 | - | 17.65s | 23.12 | - | 3.12m | - | - | - |
| Gurobi | Exact | 16.55 | | 21.81m | - | | - | - | | - |
| LKH-3 | Heuristic | 16.55 | 0.00% | 21.69s | 23.12 | 0.00% | 1.2m | 71.78 | 0.00% | 4.13m |
| Nearest Insertion | Heuristic | 20.62 | 24.59% | 0s | 28.96 | 25.26% | 0s | 90.51 | 26.11% | 0.38s |
| Random Insertion | Heuristic | 18.57 | 12.21% | 0s | 26.12 | 12.98% | 0s | 81.85 | 14.04% | 0.25s |
| Farthest Insertion | Heuristic | 18.30 | 10.57% | 0s | 25.72 | 11.25% | 0s | 80.59 | 12.29% | 0.38s |
| MMAS | Heuristic | 17.56 | 6.13% | 1.88s | 24.90 | 7.72% | 5.79s | 99.31 | 38.36% | 9.75m |
| | Heuristic+2opt | 16.57 | 0.16% | 2.49s | 23.21 | 0.42% | 4.81s | 72.64 | 1.20% | 58.58s |
| MCTS | MCTS | 16.66 | 0.66% | 50s | 23.39 | 1.16% | 100s | 74.5 | 3.79% | 16.67m |
| SoftDist | Heuristic+MCTS | 16.62 | 0.43% | 50s | 23.63 | 3.13% | 100s | 74.03 | 3.13% | 16.67m |
| AM | AR+Greedy | 21.46 | 29.71% | 0.55s | 33.55 | 45.10% | 1.49s | 153.42 | 113.75% | 8s |
| LEHD | AR+Greedy | 16.81 | 1.56% | 0.14s | 23.85 | 3.17% | 0.75s | - | - | - |
| | + Partial Re-Constriction 1000 | 16.58 | 0.17% | 33.75s | 23.29 | 0.72% | 3.28m | - | - | - |
| DeepACO | NAR+ACO | 16.86 | 1.84% | 10s | 23.85 | 3.16% | 32s | - | - | - |
| GLOP | NAR+AR | 16.91 | 1.99% | 0.7s | 23.84 | 3.11% | 1.41s | 75.29 | 4.90% | 6.75s |
| AttGCN | | 24.45 | 47.78% | 0.2s | 38.29 | 65.52% | 0.34s | 205.84 | 184.90% | 15.6s |
| DIMES | NAR+Greedy | 29.17 | 76.28% | 0.45s | 46.61 | 101.61% | 0.98s | 331.73 | 359.15% | 17.43s |
| UTSP | | 21.89 | 32.28% | 0.64s | 33.51 | 44.94% | 1.57s | - | - | - |
| DIFUSCO | | 18.35 | 10.85% | 1.69s | 26.14 | 13.06% | 5.56s | 98.15 | 36.75% | 1.78m |
| AttGCN | | 17.13 | 3.53% | 0.2s+1.45s | 24.27 | 4.97% | 0.34s+4.48s | 85.66 | 19.34% | 15.6s+6.42m |
| DIMES | **+ HEATACO** | 17.20 | 3.98% | 0.98s+1.49s | 24.32 | 5.20% | 0.98s+4.35s | 88.33 | 23.06% | 17.43s+5.51m |
| UTSP | | 17.27 | 4.39% | 0.64s+1.46s | 24.60 | 6.42% | 1.57s+4.23s | - | - | - |
| DIFUSCO | | 16.68 | 0.80% | 1.69s+1.32s | 23.40 | 1.20% | 5.56s+4.52s | 78.11 | 8.82% | 1.78m+5.31m |
| AttGCN | | 16.66 | 0.69% | 0.2s+50s | 23.37 | 1.09% | 0.34s+1.67m | 73.95 | 3.02% | 15.6s+16.67m |
| DIMES | + MCTS | 16.66 | 0.69% | 0.98s+50s | 23.37 | 1.11% | 0.98s+1.67m | 73.97 | 3.05% | 17.43s+16.67m |
| UTSP | | 16.69 | 0.90% | 0.64s+50s | 23.47 | 1.53% | 1.57s+1.67m | - | - | - |
| DIFUSCO | | 16.63 | 0.51% | 1.69s+50s | 23.24 | 0.53% | 5.56s+1.67m | 73.47 | 2.36% | 1.78m+16.67m |
| AttGCN | | 16.57 | 0.17% | 0.2s+1.83s | 23.22 | 0.44% | 0.34s+4.52s | 72.69 | 1.27% | 15.6s+1.21m |
| DIMES | **+ HEATACO** **+2opt** | 16.57 | 0.14% | 0.98s+2.04s | 23.21 | 0.39% | 0.98s+4.37s | **72.61** | **1.15%** | 17.43s+1.37m |
| UTSP | | 16.57 | 0.16% | 0.64s+1.82s | 23.22 | 0.42% | 1.57s+4.24s | - | - | - |
| DIFUSCO | | **16.56** | **0.11%** | 1.69s+2.06s | **23.17** | **0.23%** | 5.56s+4.91s | 72.63 | 1.19% | 1.78m+1.11m |

random seeds per instance. We measure per-instance wall-clock runtime on AMD EPYC 9634 CPUs with 16 threads. In all tables, a leading "+" indicates additional CPU decoding time given a fixed precomputed heatmap (heatmap inference excluded). For NAR baselines (NAR+Greedy), greedy merge overhead is negligible, so the reported runtimes largely reflect heatmap inference and are provided for context only (Appendix B.2). For reference, the published MCTS decoder uses a 128-core CPU implementation (Pan et al., 2025) with a fixed time budget of $0.1N$ seconds per instance.

### 4.5. Main results

Table 1 reports results on TSP500/TSP1K/TSP10K in terms of average tour length $L$, optimality gap (%), and per-instance runtime $T$. The table includes end-to-end solvers (exact, classical heuristics, and representative NCO methods) as references, and focuses on the decoding performance of heatmap-based NAR pipelines.

The results highlight three consistent trends. First, greedy heatmap decoding is brittle at scale: on TSP10K, AttGCN and DIMES with NAR+Greedy reach 184.90% and 359.15% gaps, despite their filtered candidate sets retaining high optimal-tour recall (Table 4 in Appendix C). This failure reflects the decoding mechanism rather than the heatmap alone: greedy merge commits to locally attractive edges and cannot reliably repair global degree/subtour conflicts, so it

is not a universal proxy for heatmap quality. Second, HEAT-ACO converts the same heatmaps into high-quality feasible tours under practical budgets. With 2-opt, gaps drop to 0.11–0.17% on TSP500, 0.23–0.44% on TSP1K, and 1.15–1.27% on TSP10K, with seconds-to-minutes additional CPU decoding time given fixed heatmaps. Third, compared to published MCTS-guided decoding, HEATACO+2opt offers a favorable quality–time trade-off on large instances: for example, on TSP10K with the DIFUSCO heatmap, +MCTS reports a 2.36% gap under a 16.67-minute decoding budget, while HEATACO+2opt reaches a 1.19% gap with 1.11 minutes of decoding time. On TSP500/1K, the advantage is even clearer: HEATACO+2opt reaches 0.11%/0.23% gaps with seconds-level decoding time, outperforming +MCTS under 50–100 second budgets without requiring bespoke action-space engineering. Appendix E shows that these trade-offs largely persist under distribution shift on TSPLIB when the heatmap ranking remains informative, but can degrade for miscalibrated heatmaps.

### 4.6. Convergence and time-to-quality

Figure 2 shows that heatmap guidance improves *time-to-quality* across scales. Compared to vanilla MMAS, HEAT-ACO drives down the excess tour length faster by biasing sampling toward the heatmap-supported candidate region while allowing pheromone feedback to suppress globally inconsistent local choices. The effect is most pronounced

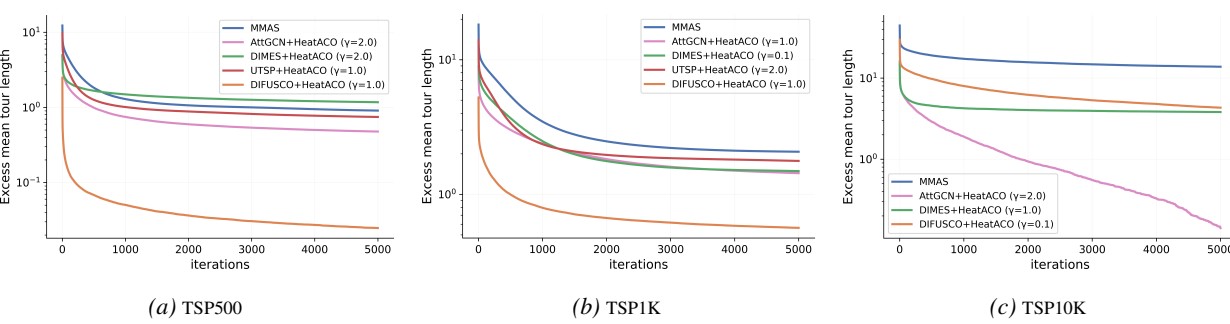

*(a) TSP500*        *(b) TSP1K*        *(c) TSP10K*

*Figure 2.* Convergence without local improvement across scales. Let $L_i(t)$ be the mean best-so-far tour length at iteration $t$ for method $i$ (averaged over instances and 10 seeds). For each panel we define a common final baseline $B = \min_i L_i(I)$ at the last iteration $I$, then plot $y_i(t) = \log_{10}(L_i(t) - B + c)$ with $c = \max(\alpha \operatorname{median}_i(L_i(t_{\mathrm{mid}}) - B), \epsilon)$, $t_{\mathrm{mid}} = 0.5I$, $\alpha = 0.03$, and $\epsilon = \max(|B| \cdot 10^{-8}, 10^{-12})$. This highlights mid/late-iteration differences without over-amplifying tiny tail gaps. Appendix F reports the corresponding curves with 2-opt local improvement.

at large scale (TSP10K), where greedy decoding is brittle and search budgets are tight. This faster convergence helps explain the strong quality–time trade-offs in Table 1.

## 5. Analysis: Heatmap Reliability and Decoding Limits

### 5.1. Sparse candidates with high optimal-tour recall

Given a predicted heatmap $H$, we build a sparse candidate set $E(H)$ via thresholding or top-$k$ filtering (Section 2.3). We measure candidate quality by its sparsity (Edges/$N$) and its tour-edge recall relative to the benchmark tour (definitions in Appendix C.1). Throughout Section 5, we refer to edges in $E(H)$ as *candidate edges*, and edges in the benchmark tour as *optimal-tour edges*. At a fixed threshold $\epsilon_h = 10^{-4}$, heatmaps yield only $O(N)$ candidate edges (typically 3–12$N$) while retaining 96–100% of optimal-tour edges (Table 4 in Appendix C). We keep $\epsilon_h$ fixed across heatmap sources for consistent comparisons (Appendix C). Thus, *candidate generation is not the main bottleneck*.

### 5.2. Tour edges concentrate in a narrow confidence band

Beyond aggregate recall, we find a consistent *confidence-mass concentration* phenomenon: Most candidate edges receive very small confidence and are rarely part of the benchmark tours, while a narrow mid/high-confidence band accounts for most of the optimal-tour recall. Concretely, the bars quantify how many candidate edges fall into each confidence interval, while the curve shows where the optimal-tour edges fall; a sharp peak indicates that most optimal-tour edges lie in a narrow confidence band. Figure 3 visualizes this effect across instance scales (Appendix D provides additional discussion). We observe the same concentration pattern on out-of-distribution real-world TSPLIB instances; see Appendix Figure 8. This concentration suggests that filtering can eliminate the bulk of low-quality edges, but also that decoders should treat $H$ as a *soft prior* and remain robust to calibration differences.

**Low-confidence collapse.** Across predictors and instances, we sometimes observe a *low-confidence collapse* pattern where a large fraction of *candidate edges* accumulates in extremely small-confidence bins yet accounts for little optimal-tour recall. In Figure 3 (e.g., DIFUSCO (Sun & Yang, 2023)), the band $[0.4, 0.5]$ contains only about $0.5N \sim 1N$ candidate edges yet covers roughly 50~80% of optimal-tour edges, whereas $[0.01, 0.05]$ can contain about $3N \sim 4.5N$ candidate edges but covers < 20%. This pattern aligns with the extreme imbalance of edge prediction (per node, tour edges vs. non-tour edges are approximately $2 : (N-2)$) and indicates that the smallest-confidence tail can become less informative. On TSPLIB, we observe a stronger version for DIMES and UTSP on pr1002/pr2392 (Appendix E.1): candidate graphs become extremely dense (e.g., Edges/$N$ in the tens to hundreds; Table 6), with many candidate edges pushed into the lowest-confidence bins. In this regime, a fixed threshold admits many low-signal candidate edges and inflates the effective branching factor, making decoding substantially harder.

### 5.3. Why Existing Decoders Struggle on Large Instances

The diagnostics above clarify why decoding remains difficult even when the candidate set has high optimal-tour recall. A heatmap is not a set of hard constraints; it is a noisy ranking over edges. After filtering, the decoder still faces two coupled tasks: enforce feasibility (degree-2, single cycle) and choose among many competing candidates per node.

**Greedy decoding commits too early.** Greedy merge (Sun & Yang, 2023) ranks edges by a local score (Eq. (7)) and commits to choices that satisfy only local feasibility checks (degree $\leq 2$ and basic subtour avoidance). When multiple nodes share many competing high-confidence neighbours —a typical situation when $|E| \approx cN$ with $c$ in the single digits—early decisions create degree conflicts and force long detours later. Crucially, repairing a single wrong edge can require coordinated changes across distant parts of the partial tour, which a greedy procedure does not explore.

**MCTS-guided improvement is powerful but costly.** MCTS-guided $k$-opt (Fu et al., 2021) attempts to avoid early commitment by exploring sequences of local improvements. However, the branching factor grows quickly with $N$; tree

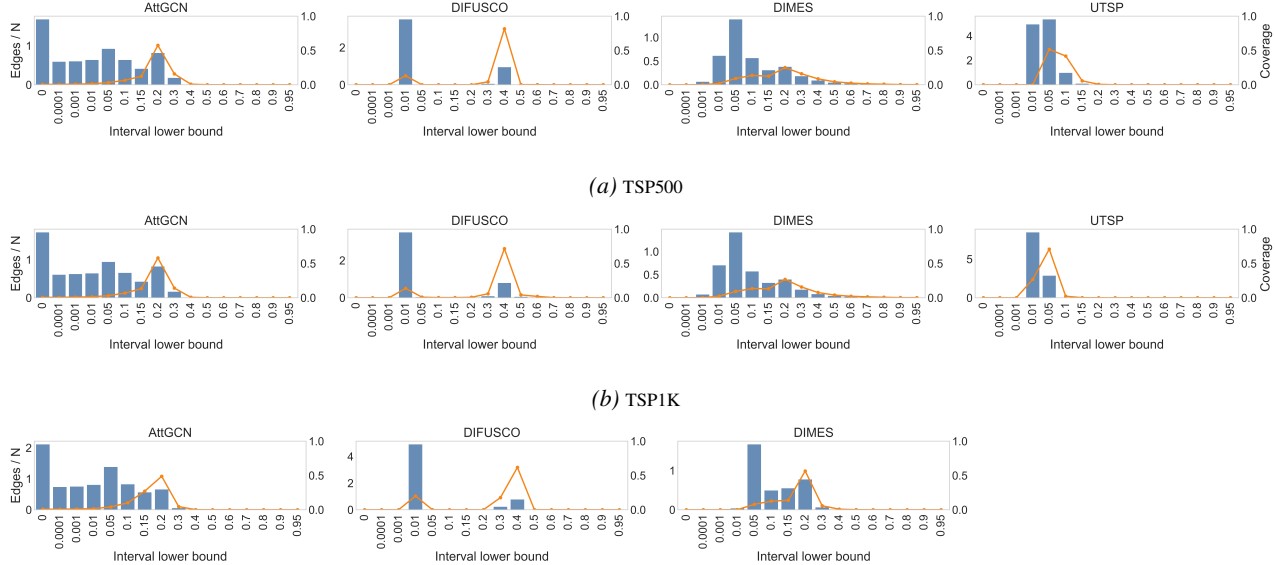

*(a)* TSP500

*(b)* TSP1K

*(c)* TSP10K

*Figure 3.* Interval contribution analysis of heatmap confidence values across instance scales. We partition heatmap scores into confidence intervals on $[0, 1]$. Bars report the (normalised) number of candidate edges per node whose confidence falls in each interval. The curve is normalised over the optimal-tour edges: for each interval, it reports the fraction of optimal-tour edges whose confidence falls in that interval (so the curve sums to 1 across intervals). Benchmark-tour edges concentrate in a narrow mid/high-confidence band, while the low-confidence tail contains many candidate edges but very few optimal-tour edges.

expansion introduces sequential dependencies; and performance is sensitive to move proposal and exploration settings. As a result, strong performance on large instances often requires large time budgets and substantial engineering. In the released parallel MCTS decoder we compare against (Pan et al., 2025), preprocessing computes distance/heatmap matrices in $\mathcal{O}(N^2)$ time and builds candidate sets in $\mathcal{O}(N^2 k)$ time (candidate size $k$). Each MCTS simulation samples $\mathcal{O}(c_{\text{sim}} N)$ actions (for a constant sampling factor $c_{\text{sim}}$) to a maximum depth $D$, yielding $\mathcal{O}(c_{\text{sim}} D N (N+k))$ work per simulation, and the overall runtime is controlled by the wall-clock budget ($0.1N$ seconds in our comparisons); Appendix B.3 provides a code-level breakdown.

**Implication for decoder design.** These failure modes motivate decoders that (i) treat the heatmap as a *soft prior* rather than a hard filter, (ii) incorporate *global feedback* to correct locally attractive mistakes, and (iii) scale efficiently on sparse candidates. HEATACO instantiates this design by injecting the heatmap prior into MMAS sampling while using pheromone updates as global feedback (Section 3).

## 6. Ablations and Sensitivity

### 6.1. Ablation: heatmap prior vs. distance heuristic

We ablate the two guidance terms in Eq. (9). Setting $\gamma = 0$ removes heatmap guidance and recovers standard MMAS; setting $\beta = 0$ removes the distance heuristic and tests how far decoding can go using only the heatmap prior and pheromone feedback. Table 2 reports gaps across heatmaps.

*Table 2.* Component ablations of HEATACO. We report optimality gap (%) on TSP500/1K/10K. "$\gamma{=}0$" removes heatmap guidance; "$\beta{=}0$" removes the distance heuristic.

| Variant | Heatmap | TSP500 | TSP1K | TSP10K |
|---|---|---|---|---|
| HEATACO | AttGCN | 3.53% | 4.97% | 19.34% |
| | DIMES | 3.98% | 5.20% | 23.06% |
| | UTSP | 4.39% | 6.42% | - |
| | DIFUSCO | 0.80% | 1.20% | 8.82% |
| MMAS ($\gamma{=}0$) | - | 6.13% | 7.72% | 38.36% |
| HEATACO ($\beta{=}0$) | AttGCN | 7.35% | 10.00% | 54.90% |
| | DIMES | 11.40% | 18.81% | 34.3% |
| | UTSP | 12.78% | 87.26% | - |
| | DIFUSCO | 0.92% | 1.52% | 8.92% |
| HEATACO +2opt | AttGCN | 0.17% | 0.44% | 1.27% |
| | DIMES | 0.14% | 0.39% | 1.15% |
| | UTSP | 0.16% | 0.42% | - |
| | DIFUSCO | 0.11% | 0.23% | 1.19% |
| MMAS +2opt ($\gamma{=}0$) | - | 0.16% | 0.42% | 1.20% |
| HEATACO +2opt ($\beta{=}0$) | AttGCN | 0.24% | 0.72% | 5.84% |
| | DIMES | 1.76% | 3.90% | 7.06% |
| | UTSP | 0.15% | 0.23% | - |
| | DIFUSCO | 0.13% | 0.24% | 1.71% |

Two patterns emerge. Without local improvement, the heatmap prior provides a consistent boost over vanilla MMAS (compare HEATACO vs. $\gamma{=}0$), confirming that heatmaps contain actionable structural information even after simple filtering. The distance heuristic remains important on the hardest regimes (notably TSP10K), but the $\beta{=}0$ ablation can remain competitive for some heatmaps (e.g., DIFUSCO on TSP500/1K), indicating that pheromone feedback can correct part of the heatmap noise. With 2-opt

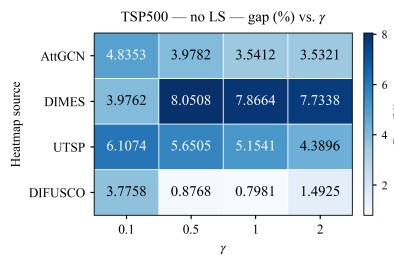 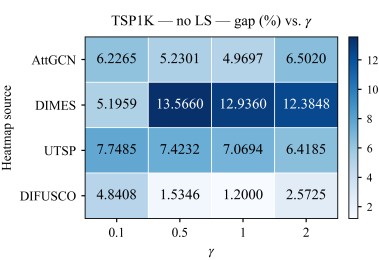 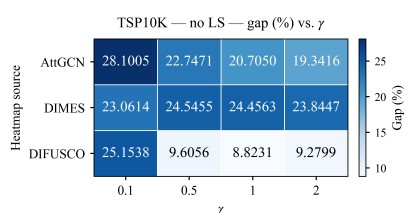

*Figure 4.* Sensitivity of HEATACO (no local search) to the heatmap exponent $\gamma$ across instance scales. Each cell reports the optimality gap (%) achieved by HEATACO when using a fixed heatmap source and a coarse $\gamma \in \{0.1, 0.5, 1.0, 2.0\}$ sweep.

*Table 3.* 3-opt local improvement results. We report optimality gap (%) and per-instance runtime; runtime reporting follows Section 4.4. UTSP does not provide TSP10K heatmaps.

| | TSP 500 | | |
|---|---|---|---|
| Method | Type | Gap | Time |
| MMAS | Heuristic+3opt | **0.003%** | 7.5s |
| AttGCN | | 0.006% | + 6.84s |
| DIMES | **+ HEATACO** | 0.004% | + 8.05s |
| UTSP | **+3opt** | 0.005% | + 6.97s |
| DIFUSCO | | 0.006% | + 6.51s |
| | TSP 1K | | |
| MMAS | Heuristic+3opt | 0.06% | 5.79s |
| AttGCN | | 0.07% | + 14.60s |
| DIMES | **+ HEATACO** | 0.06% | + 15.08s |
| UTSP | **+3opt** | 0.06% | + 15.06s |
| DIFUSCO | | **0.05%** | + 15.63s |
| | TSP 10K | | |
| MMAS | Heuristic+3opt | 0.41% | 3.23m |
| AttGCN | | 0.44% | + 4.39m |
| DIMES | **+ HEATACO** | **0.40%** | + 3.73m |
| DIFUSCO | **+3opt** | 0.43% | + 4.28m |

enabled, the performance gap between heatmap-guided and heatmap-free MMAS narrows substantially, suggesting that local search can repair many residual conflicts. Stronger 3-opt can further tighten solutions at additional compute cost; see Section 6.3.

### 6.2. Sensitivity to the heatmap exponent $\gamma$

Figure 4 studies the heatmap exponent $\gamma$ in Eq. (9), which controls how strongly HEATACO follows a fixed heatmap via $\tilde{H}_{ij}^{\gamma}$. Larger $\gamma$ concentrates probability mass on high-confidence edges (lower-entropy sampling), while smaller $\gamma$ softens the prior and relies more on pheromone feedback and the distance heuristic. Since lightly filtered heatmaps already retain high tour-edge recall (Appendix C), tuning $\gamma$ is primarily a *calibration* choice: it trades off trusting the predicted confidence band versus keeping enough exploration to resolve global degree/subtour conflicts. Empirically, the best $\gamma$ depends on both the heatmap source and the instance scale: stronger guidance can help on large instances by reducing effective branching, but overly large $\gamma$ may overtrust miscalibrated edges and trap the search. In practice, a coarse sweep $\gamma \in \{0.1, 0.5, 1.0, 2.0\}$ is sufficient, and Appendix G.1 provides additional discussion and the corresponding sensitivity when 2-opt/3-opt local improvement

is enabled. When benchmark tours are unavailable, Appendix G.2 provides a label-free entropy-targeted rule to select $\gamma$ from the heatmap scores alone.

### 6.3. Stronger local improvement (3-opt)

Table 3 evaluates a stronger local improvement operator (3-opt). With 3-opt, both MMAS and HEATACO approach the benchmark tours on TSP500/1K. On TSP10K, HEATACO+3opt reaches gaps around 0.40–0.44% (depending on the heatmap source), at the cost of several minutes of additional decoding time. This provides a clear quality–time knob: 2-opt is a strong default for minute-level budgets, while 3-opt is useful when additional compute is available.

## 7. Conclusions

Heatmap-based non-autoregressive TSP solvers shift the computational burden from prediction to decoding: a feasible tour must satisfy global degree and subtour constraints, and naive greedy merging can fail catastrophically at large scale. This work focuses strictly on the heatmap-to-tour decoding stage. We proposed HEATACO, a simple and modular MMAS-based decoder that treats the heatmap as a soft prior and uses pheromone feedback as instance-specific global conflict resolution, with optional 2-opt/3-opt refinement. Across heatmaps inferred by multiple pretrained predictors, HEATACO achieves strong quality–time trade-offs on TSP500/1K/10K under practical CPU budgets given fixed heatmaps, and improves time-to-quality relative to vanilla MMAS by leveraging the heatmap prior when it is informative.

Our analysis also clarifies the key limitation: decoding quality is ultimately bounded by heatmap reliability. We do not claim that HEATACO uniformly dominates vanilla MMAS in all settings; rather, its benefit depends on how well the heatmap preserves a useful edge ranking after filtering. Under distribution shift, miscalibration and low-confidence collapse can weaken this ranking, reduce the benefit of heatmap guidance, and bring performance closer to heatmap-free decoding. We therefore view improving heatmap calibration and out-of-distribution generalisation as a direct and promising path to further strengthening HEATACO-style decoding.

## Impact Statement

This work advances machine learning for combinatorial optimisation by improving the decoding stage of heatmap-based TSP solvers. A more reliable decoder can benefit route-planning workflows in logistics and operations research by improving tour quality with low computational overhead. As with many optimisation techniques, it could also be misused to increase efficiency in harmful contexts (e.g., military or illicit logistics). Our experiments use standard public benchmarks and do not involve personal data. We will open-source the implementation to support reproducibility. We encourage responsible deployment in domain-specific settings, including risk assessment, incorporating safety/legal/social constraints beyond distance minimisation, and maintaining human oversight in high-stakes applications. The method is lightweight (minutes wall-clock in our experiments), which lowers environmental footprint but may also lower the barrier to misuse; practitioners should consider access controls and appropriate governance when deploying in sensitive contexts.

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

**Appendix overview.** The appendix is organised to directly support the main paper:

- **Appendix A**: reproducibility snapshot (notation, candidate sets, and runtime accounting).

- **Appendix B**: implementation and complexity details for MMAS/HEATACO, plus baseline provenance (Appendix B.2) and MCTS complexity (Appendix B.3).

- **Appendix C**: heatmap-reliability diagnostics used in Section 5.

- **Appendix D**: additional discussion of confidence-mass patterns supporting Section 5.2.

- **Appendix E**: TSPLIB out-of-distribution evaluation and diagnostics (supporting Sections 4.1 and 5).

- **Appendix F**: additional convergence plots with local improvement.

- **Appendix G**: additional ablations and sensitivity analyses.

- **Appendix H**: extended related work.

## A. Reproducibility Snapshot

**Connection to the main paper.** This section provides a minimal, self-contained summary of notation and experimental conventions for readers who jump directly between the main paper and the appendix.

**Notation sanity check.** Throughout the paper, we use:

- heatmap $H \in [0,1]^{N \times N}$ (edge confidence) and pheromone $\tau \in \mathbb{R}_+^{N \times N}$ (MMAS);

- heatmap threshold $\epsilon_h$ (used for diagnostics and preprocessing), and floor value $\varepsilon$ used in the transition rule;

- per-node candidate-list size $k$ (default $k{=}20$), ants $m$, iterations $I$, and heatmap exponent $\gamma$ in Eq. (9).

**Raw filtered graph versus decoder candidate lists.** We use two related but distinct candidate structures:

- **Raw filtered graph (diagnostics).** We define the thresholded edge set $E_{\epsilon_h}(H) = \{(i,j) \mid H_{ij} \geq \epsilon_h\}$ (Appendix C.1) and report Edges/$N$, Coverage, and Miss on this set (Tables 4 and 6). This set is *not* capped per node and can become dense under distribution shift, which is a useful signal for heatmap miscalibration (Section 5).

- **Decoder candidate lists (decoding).** HEATACO constructs fixed-size per-node lists of size $k$ by taking the top-ranked heatmap neighbours above $\epsilon_h$ and filling any remaining slots with nearest neighbours by distance (Appendix B). Decoding and local improvement are restricted to these lists to ensure $O(Nk)$ neighbourhood search.

This separation resolves the apparent discrepancy that TSPLIB heatmaps can yield very large Edges/$N$ in diagnostics (e.g., for DIMES), while decoding still operates on $k{=}20$ candidates per node.

**Runtime accounting.** In all tables, a leading "+" indicates additional CPU decoding time given a fixed precomputed heatmap (heatmap inference excluded). For greedy decoding (NAR+G), greedy-merge overhead is negligible, so the reported runtimes largely reflect heatmap inference and are provided for context only. For MCTS-guided decoding, runtimes follow the published parallel implementation with a fixed time budget of $0.1N$ seconds on a 128-core CPU (Pan et al., 2025) (details in Appendix B.3).

## B. Implementation and Experimental Setup

This appendix summarises key implementation choices for MMAS/HEATACO and complements the experimental protocol in Section 4.4. We place low-level engineering details here to keep the main paper focused on the decoding formulation and key hyperparameters.

**Implementation.** We use a multi-threaded C++ implementation and run experiments on AMD EPYC 9634 CPUs with 16 threads unless noted otherwise.

**Heatmap preprocessing.** We symmetrize heatmaps (Eq. (5)), then apply a small confidence threshold $\epsilon_h = 10^{-4}$ (Section 3.2). In the transition rule, entries below $\epsilon_h$ are assigned a small floor value $\varepsilon = 10^{-9}$ to avoid zero-probability moves when distance-based fallback edges are used.

**Candidate lists (used for decoding).** We use per-node candidate lists of size $k{=}20$. For HEATACO, we seed each list with the top-ranked heatmap neighbours above $\epsilon_h$ and, if fewer than $k$ remain, fill the list with the nearest neighbours by distance.

**Transition rule interpretation.** Eq. (9) combines three signals: (i) the geometric heuristic $\eta_{ij}$ encourages short edges; (ii) pheromone $\tau_{ij}$ accumulates global evidence from previously sampled tours; and (iii) the neural prior $H_{ij}$ focuses sampling on edges that are structurally plausible under the predictor. Pheromone feedback provides a lightweight

mechanism for conflict resolution: if a high-confidence edge repeatedly leads to tours with long detours or bad sub-tour structure, its pheromone value is suppressed relative to edges that participate in shorter globally consistent tours.

**Search budget note.** Our evaluation protocol (iteration budget, seeds, and CPU) is described in Section 4.4. For context, the original MMAS work commonly uses 25 ants and a termination criterion of constructing $2,500N$ tours (where $N$ is the instance size), which exceeds our fixed iteration budget on all scales and thus makes MMAS/HEATACO comparisons conservative.

**Local improvement.** When enabled, 2-opt/3-opt moves are restricted to the filtered candidate graph (heatmap-induced candidates with KNN augmentation).

## B.1. Complexity of MMAS/HEATACO

**Notation.** Let $N$ be the number of nodes, $m$ the number of ants, $I$ the number of iterations, and $k$ the per-node candidate-list size (default $k=20$). When local improvement is enabled, let $p_2$ and $p_3$ denote the maximum number of 2-opt/3-opt passes, respectively.

**Preprocessing (one-time).** We precompute the distance matrix in $\mathcal{O}(N^2)$ time and memory. Candidate lists are constructed by (i) scanning the heatmap rows and selecting top-ranked neighbours and (ii) filling the remainder by $k$ nearest neighbours in distance. This costs $\mathcal{O}(N^2)$ for scanning plus $\mathcal{O}(Nk \log k)$ for per-row top-$k$ maintenance/sorting.

**Per-iteration tour construction.** Each ant constructs a tour of length $N$, and each step considers at most $k$ candidate next nodes. This yields $\mathcal{O}(mNk)$ time per iteration.

**Local improvement (optional).** When enabled, 2-opt restricted to the candidate graph costs $\mathcal{O}(mp_2Nk)$. Similarly, 3-opt restricted to the candidate graph costs approximately $\mathcal{O}(mp_3Nk^2)$, as each move evaluates combinations of candidate edges.

**Pheromone updates.** With candidate-based evaporation, pheromone updates and auxiliary statistics scale as $\mathcal{O}(Nk)$ per iteration. Without candidate restriction (used in our no-local-search setting), evaporation over the full matrix costs $\mathcal{O}(N^2)$ per iteration.

**Total time and memory.** With local improvement (candidate-based evaporation), the total time is

$$\mathcal{O}(N^2) \;+\; I \cdot \big(\mathcal{O}(mNk) + \mathcal{O}(\text{LS}) + \mathcal{O}(Nk)\big),$$

where $\mathcal{O}(\text{LS})$ is $\mathcal{O}(mp_2Nk)$ for 2-opt or $\mathcal{O}(mp_3Nk^2)$ for 3-opt. Without local improvement (full-matrix evaporation),

the total time is

$$\mathcal{O}(N^2) \;+\; I \cdot \big(\mathcal{O}(mNk) + \mathcal{O}(N^2)\big).$$

Memory is dominated by the distance and pheromone matrices, $\mathcal{O}(N^2)$, plus candidate lists and local caches, $\mathcal{O}(Nk)$.

## B.2. Experimental setup and provenance

**Benchmarks and references.** We use the standard AttGCN TSP benchmarks (Fu et al., 2021) (TSP500/1K/10K). For TSP500/1K, the benchmark tour length $L^\star$ corresponds to optimal tours (computed by Concorde). For TSP10K, $L^\star$ is the best-known tour released with the benchmark (computed by LKH-3). Unless otherwise stated, we compute optimality gaps relative to these references.

**Heatmap provenance.** We obtain heatmaps by running released pretrained predictors from AttGCN, DIMES, UTSP, and DIFUSCO on the benchmark instances. We do not retrain predictors; all results isolate the decoding stage given fixed heatmaps. Heatmaps are symmetrised and lightly filtered as described in Appendix B.

**Baseline provenance and runtime reporting.** The "NAR+G" rows in Table 1 are computed by running the DIFUSCO-released greedy decoder on the SoftDist-released heatmaps (Xia et al., 2024). The "+MCTS" and "SoftDist" rows follow the published parallel MCTS-guided $k$-opt decoder (Pan et al., 2025), which is tuned with per-heatmap and per-scale hyperparameters; we use the released configurations for each heatmap source and problem size. Our MMAS/HEATACO runtimes follow Section 4.4. For baselines copied from prior work (see Table 1 and its caption), we convert reported totals to per-instance time when possible; due to differences in hardware, tuning, and implementation, these numbers should be interpreted as indicative rather than strictly comparable.

## B.3. Complexity of the published MCTS decoder

We summarize the time and memory complexity of the published parallel MCTS-guided $k$-opt decoder based on its released implementation (Pan et al., 2025). This analysis follows the code structure and does not require a benchmark tour. Let $N$ be the number of nodes; $k$ be the per-node candidate-list size (upper bounded in the code by a constant max); $D$ be the maximum MCTS depth (a small constant); $c_{\text{sim}}$ be the per-round sampling factor; and $b$ be the time-budget coefficient (so the budget is $bN$ seconds; $b=0.1$ in our comparisons).

**Preprocessing.** Computing the all-pairs distance matrix and reading the heatmap both take $\mathcal{O}(N^2)$ time and mem-

ory. Candidate construction repeatedly selects the best uns-elected city by scanning all nodes, which costs $\mathcal{O}(N)$ per selection. Building $k$ candidates for each of the $N$ nodes therefore takes $\mathcal{O}(N^2 k)$ time in the worst case (and becomes $\mathcal{O}(N^3)$ if $k = \Theta(N)$).

**2-opt local search.** The implementation checks candidate neighbours for each node and may trigger path reversal operations. Let $I_{2\mathrm{opt}}$ be the number of accepted 2-opt improvements; the resulting cost is $\mathcal{O}(I_{2\mathrm{opt}} N(k+1))$, which is typically far smaller than $\mathcal{O}(N^2)$ in practice.

**MCTS rollouts under a time budget.** The dominant operations in a simulated action scale as $\mathcal{O}(D(N+k))$. Each simulation samples $\mathcal{O}(c_{\mathrm{sim}} N)$ actions, giving $\mathcal{O}(c_{\mathrm{sim}} N D(N+k))$ per simulation. The outer loop is controlled by a wall-clock budget of $bN$ seconds in the released evaluation, so total computation is best expressed in terms of throughput: with $I_{\mathrm{MCTS}}$ simulations, the total work is $\mathcal{O}(I_{\mathrm{MCTS}} c_{\mathrm{sim}} DN(N+k))$.

**Memory.** The implementation maintains multiple $N \times N$ matrices (e.g., distances and heatmap-derived weights), requiring $\mathcal{O}(N^2)$ memory; candidate lists add $\mathcal{O}(Nk)$.

**Summary.** Overall, the decoder has $\mathcal{O}(N^2 + N^2 k)$ pre-processing and $\mathcal{O}(N^2)$ memory, and its time-to-quality is determined by the time-budgeted MCTS rollouts.

# C. Candidate-Set Construction and Diagnostics

**Connection to the main paper.** This appendix defines the *raw filtered-graph* diagnostics used in Section 5 to characterise heatmap reliability. Decoder-side per-node candidate lists (used by HEATACO for $\mathcal{O}(Nk)$ decoding) are described in Appendix B.

## C.1. Metrics for candidate quality

For each instance, let $E^\star \subset E_{\mathrm{full}}$ be the edge set of a *benchmark tour* (optimal when available, otherwise best-known). Given a predicted heatmap $H$ and a filtering rule, we obtain a candidate edge set $E(H)$ such as the *raw thresholded graph* $E_{\epsilon_h}(H)$ in Eq. (10) or a capped row-wise variant $E_k(H)$ in Eq. (11). We summarise candidate quality along two axes: *sparsity* (how many candidate edges remain) and *optimal-tour recall* (how many optimal-tour edges are preserved). Unless stated otherwise, Tables 4 and 6 report statistics on the raw filtered graph $E_{\epsilon_h}(H)$ *without* any per-node cap, to expose densification/collapse effects under distribution shift (Appendix A).

**Candidate-edge filtering rules.**

$$E_{\epsilon_h}(H) \triangleq \{(i,j) \in E_{\mathrm{full}} \mid H_{ij} \geq \epsilon_h\}, \qquad (10)$$

$$E_k(H) \triangleq \bigcup_{i \in V} \{(i,j) \mid j \in \text{Top-}k(H_{i\cdot})\}, \qquad (11)$$

where Top-$k(\cdot)$ selects the $k$ largest entries in a row and $V$ is the node set.

**Candidate size.**

$$M(H) \triangleq |E(H)|. \qquad (12)$$

Since the complete graph has $|E_{\mathrm{full}}| = O(N^2)$ edges, an effective filter should yield $M(H) = O(N)$ to enable scalable decoding. For readability we report $\text{Edges}/N \triangleq M(H)/N$, i.e., the number of candidate undirected edges per node.

**Edge recall of the benchmark tour.**

$$\text{Cov}(H) \triangleq \frac{|E(H) \cap E^\star|}{|E^\star|} = \frac{1}{N}|E(H) \cap E^\star|, \qquad (13)$$

because a tour contains exactly $|E^\star| = N$ undirected edges. We also report the number of missing optimal-tour edges,

$$\text{Miss}(H) \triangleq |E^\star \setminus E(H)| = N - |E(H) \cap E^\star|, \qquad (14)$$

and $\text{Miss}(H)/N$ as a percentage.

**Interpreting Table 4.** Table 4 compares sparsity and tour-edge miss rate at a fixed threshold $\epsilon_h = 10^{-4}$. Near-perfect tour-edge recall is achievable with $O(N)$ candidates using simple geometric graphs (e.g., KNN20) and is also attainable for modern heatmaps, albeit with calibration-dependent trade-offs between sparsity and miss rate.

**Threshold choice ($\epsilon_h = 10^{-4}$).** We use $\epsilon_h = 10^{-4}$ following the standard heatmap filtering used in the released AttGCN MCTS decoder implementation; several subsequent heatmap-based pipelines reuse the same setting. Fixing $\epsilon_h$ enables consistent comparisons of candidate sparsity/recall across heatmap sources.

**Discussion.** Table 4 confirms that a single global threshold can reduce dense heatmaps to sparse $\mathcal{O}(N)$ candidate graphs, turning decoding from an $O(N^2)$ edge selection problem into a constrained selection problem on a sparse but noisy graph. However, sparsity and recall trade off in a calibration-dependent way. For example, DIFUSCO yields about 4.5–5.9 edges/$N$ with $< 1\%$ miss across scales, while DIMES is slightly sparser (about 3.6–3.8 edges/$N$) but misses 2.5–3.4% of optimal-tour edges. At $N=10^4$, a 1% miss corresponds to roughly 100 optimal-tour edges absent from $E(H)$, which can reduce the reliability of $k$-opt style improvement restricted to the candidate graph

*Table 4.* Candidate sparsity, tour-edge recall, and auxiliary cross-entropy diagnostics at $\epsilon_h = 10^{-4}$. Edges/$N$ reports $|E_{\epsilon_h}(H)|/N$ on the *raw* thresholded graph (no per-node cap; lower is sparser). Missing (%) reports $100 \cdot \text{Miss}(H)/N$ (lower is better). Coverage (%) is $100 \cdot \text{Cov}(H)$, and Missing Edges reports $\text{Miss}(H)$. For heatmap methods we also report the binary cross entropy (CE) and class-weighted cross entropy (WCE) of the original heatmap against the optimal-tour adjacency, as an auxiliary, label-based proxy for confidence calibration and ranking (lower indicates better alignment with the reference labels). CE/WCE are not defined for deterministic geometric graphs (KNN-$k$).

| Method | N | Edges/$N$ | Missing (%) | Coverage (%) | Miss (edges) | CE | WCE |
|---|---|---|---|---|---|---|---|
| KNN5 | 500 | 3.00 | 2.0359% | 97.9641% | 10.18 | - | - |
| | 1K | 2.99 | 1.8875% | 98.1125% | 18.88 | - | - |
| | 10K | 2.98 | 2.2212% | 97.7788% | 222.12 | - | - |
| KNN10 | 500 | 5.78 | 0.0703% | 99.9297% | 0.35 | - | - |
| | 1K | 5.75 | 0.0680% | 99.9320% | 0.68 | - | - |
| | 10K | 5.71 | 0.1337% | 99.8663% | 13.37 | - | - |
| KNN15 | 500 | 8.55 | 0.0047% | 99.9953% | 0.02 | - | - |
| | 1K | 8.48 | 0.0047% | 99.9953% | 0.05 | - | - |
| | 10K | 8.39 | 0.0094% | 99.9906% | 0.94 | - | - |
| KNN20 | 500 | 11.32 | 0.0016% | 99.9984% | 0.01 | - | - |
| | 1K | 11.20 | 0% | 100% | 0 | - | - |
| | 10K | 11.05 | 0% | 100% | 0 | - | - |
| DIFUSCO | 500 | 4.51 | 0.6219% | 99.3781% | 3.11 | **0.00608** | **0.69283** |
| | 1K | 4.51 | 0.6203% | 99.3797% | 6.20 | **0.00302** | **0.69044** |
| | 10K | 5.90 | 0.3281% | 99.6719% | 32.81 | **0.00036** | **0.79809** |
| AttGCN | 500 | 4.78 | 0.8359% | 99.1641% | 4.18 | 0.00788 | 0.83959 |
| | 1K | 4.86 | 0.8898% | 99.1102% | 8.90 | 0.00400 | 0.85204 |
| | 10K | 5.76 | 1.5737% | 98.4263% | 157.37 | 0.00046 | 0.97910 |
| DIMES | 500 | 3.67 | 3.3969% | 96.6031% | 16.98 | 0.00966 | 1.07837 |
| | 1K | 3.82 | 3.0594% | 96.9406% | 30.59 | 0.00475 | 1.05666 |
| | 10K | 3.58 | 2.5175% | 97.4825% | 251.75 | 0.00049 | 1.06126 |
| UTSP | 500 | 11.39 | 0.0016% | 99.9984% | 0.01 | 0.01145 | 1.14546 |
| | 1K | 11.34 | 0.0008% | 99.9992% | 0.01 | 0.00651 | 1.42808 |
| | 10K | - | - | - | - | - | - |

(although miss is only a diagnostic, since near-optimal tours need not coincide exactly with the benchmark tour). Conversely, UTSP behaves closer to a geometric KNN graph: it is denser (about 11.3 edges/$N$) yet nearly lossless in recall, improving robustness but increasing the number of competing edges that a decoder must disambiguate. Overall, these diagnostics reinforce the main paper's conclusion that large-scale decoding is less about "recovering missing edges" and more about managing false positives and enforcing global consistency under degree/subtour constraints. For out-of-distribution diagnostics on TSPLIB instances, see Appendix E.

### C.2. Auxiliary CE/WCE diagnostics

**Calibration-aware diagnostics via cross entropy.** To contextualise heatmap quality beyond thresholded recall, we compute the binary cross entropy (CE) and the class-weighted cross entropy (WCE) of the *original* heatmap against the optimal-tour adjacency (Appendix C.1). CE/WCE are commonly used training objectives and provide an auxiliary, label-based proxy for how well a predictor separates optimal-tour edges from non-tour edges under the extreme class imbalance of edge prediction. Figure 5 shows that higher CE/WCE often coincide with worse decoding gaps, and that strong decoders (HEATACO+2opt/3opt) are substantially more robust than greedy merge but remain

bounded by heatmap reliability. However, CE/WCE are only auxiliary indicators: decoding depends on the *relative* ranking of edges and global feasibility, and similar CE/WCE values can arise from different calibration patterns that affect decoders differently. Under distribution shift, CE/WCE and candidate sparsity can change sharply (Table 6 in Appendix E.1), aligning with TSPLIB decoding failures for some predictors (Appendix E).

**Implication for unlabeled settings.** CE/WCE require a benchmark tour and are therefore most useful for benchmarking or curated test sets (including TSPLIB, which provides best-known tours). In unlabeled deployment scenarios, a practical proxy is to monitor label-free quantities such as Edges/$N$ and the confidence-mass profile (Section 5.2): extreme densification or a collapsed low-confidence tail signals that the heatmap prior is less informative and that stronger exploration (smaller $\gamma$) and local improvement may be required. In addition, since the decoder directly optimises tour length, $\gamma$ can be selected without labels by a short coarse sweep (e.g., $\{0.1, 0.5, 1, 2\}$) with a small iteration budget and choosing the best tour; Section 6.2 shows that such coarse tuning is sufficient in practice. As an even cheaper alternative, Appendix G.2 introduces an entropy-targeted rule that selects $\gamma$ from the heatmap scores alone, without running decoding or requiring benchmark tours.

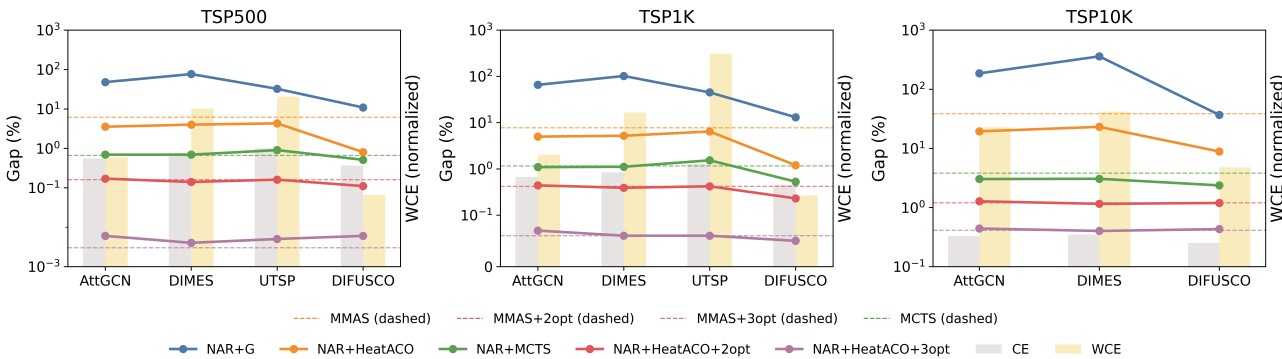

*Figure 5.* Decoding gap versus auxiliary heatmap cross-entropy diagnostics on TSP500/1K/10K benchmarks. Lines show the optimality gap (%) achieved by different decoders for each heatmap source; bars show the corresponding CE and WCE values from Table 4 (normalised per instance scale for visualisation). In the legend, "NAR+ACO" denotes HEATACO (heatmap-guided MMAS), and "+2opt/+3opt" denote adding local improvement.

## D. Additional Discussion: Confidence-Mass Diagnostics

**Connection to the main paper.** This section expands the confidence-band discussion in Section 5.2 and highlights implications for decoder robustness.

**Importance of confidence mass.** Beyond aggregate recall, Figure 3 shows that modern predictors allocate confidence very unevenly: a narrow mid/high-confidence band contains most of the optimal-tour edge signal, while the vast majority of candidate edges sit in a low-confidence region that contains almost no optimal-tour edges. This partially explains why greedy merge can fail severely at scale: early choices are made under degree/subtour constraints with only local feasibility checks, and mistakes require coordinated global changes that greedy decoding does not explore.

**Relation to decoder robustness.** Section 5.2 discusses the low-confidence collapse pattern and its implications for decoding. Here we highlight an additional practical point: although confidence mass concentrates across predictors and instance sizes, the sharpness and location of the mid/high-confidence band can vary, motivating conservative use of the heatmap as a *relative* prior and coarse tuning of the heatmap strength $\gamma$ (Appendix G.1).

For confidence-mass diagnostics under distribution shift on TSPLIB, see Appendix E.

## E. TSPLIB Generalisation and Diagnostics

**Connection to the main paper.** This appendix complements the TSPLIB discussion in Sections 4.1 and 5 by reporting full OOD results and diagnostics.

Table 5 reports out-of-distribution results on three TSPLIB instances (Reinelt, 1991). We generate heatmaps (and greedy NAR+G decoding) using the released author code and pretrained models. Consistent with Section 4.4, we exclude heatmap inference time from runtime reporting; our GPU is NVIDIA RTX 2000 Ada, so inference times are not directly comparable across sources. MCTS results are taken from the original paper (Pan et al., 2025). For candidate-set diagnostics on the same instances, see Table 6 and Figure 8.

*(a)* TSP500 (uniform)    *(b)* TSP1K (uniform)    *(c)* TSP10K (uniform)

*(d)* pcb442 (PCB)    *(e)* pr1002 (drilling)    *(f)* pr2392 (drilling)

*Figure 6.* Instance geometry shift from standard random benchmark instances (top row) to real-world TSPLIB instances (bottom row). TSPLIB layouts are more structured and contain many locally similar configurations, which makes confidence calibration and downstream decoding more challenging.

**TSPLIB as a substantial distribution shift.** The standard TSP500/1K/10K benchmarks draw node coordinates i.i.d. from a uniform distribution in the unit square, so local neighbourhoods are "generic" and geometric cues are relatively distinct. In contrast, TSPLIB instances are derived from real applications and often exhibit strong geometric regularities (e.g., circuit-board layouts and drilling patterns), which create many locally similar or near-equidistant alternatives and amplify the risk of miscalibrated confidence. Figure 6 visualizes this contrast for the three TSPLIB instances used in Table 5.

**Results and discussion.** Greedy decoding (NAR+G) exhibits poor robustness under distribution shift, yielding substantial gaps despite using the same pretrained predictors. HEATACO converts the same fixed heatmaps into substantially better tours within seconds, and with a lightweight 2-opt pass it reaches sub-1% gaps for UTSP/DIFUSCO (and ≈0.8% for AttGCN) on all three instances while remaining in the seconds regime. Stronger 3-opt fur-

ther tightens gaps—often by approximately one order of magnitude—while remaining in the seconds regime: on pcb442, HEATACO+3opt matches the TSPLIB reference for UTSP/DIMES and is within 0.0014% for AttGCN/DIFUSCO; on pr1002 and pr2392 it achieves gaps as low as 0.0333% and 0.1960%, respectively. Compared to the published parallel MCTS decoder (tens of seconds to minutes), HEATACO+3opt achieves comparable or better gaps for AttGCN/UTSP/DIFUSCO at 1–2 orders of magnitude lower decoding time, without extensive action-space engineering.

### E.1. Candidate diagnostics under distribution shift

**Discussion.** Table 6 shows that distribution shift can substantially change candidate-set statistics even under the same filtering threshold $\epsilon_h$. DIMES and UTSP exhibit a pronounced *low-confidence collapse* on pr1002/pr2392: the filtered candidate graphs become extremely dense (tens to hundreds of edges per node) and many edges are pushed

*Table 5.* Generalization to out-of-distribution TSPLIB instances (Reinelt, 1991). We report tour length, optimality gap (%, lower is better) to the TSPLIB reference, and per-instance runtime. Times are in seconds unless marked with "m"; "−" indicates not reported. "+" indicates additional decoding time given a fixed heatmap (Section 4.4).

| Method | Type | pcb442 | | | pr1002 | | | pr2392 | | |
|---|---|---|---|---|---|---|---|---|---|---|
| | | Length | Gap (%) | Time | Length | Gap (%) | Time | Length | Gap (%) | Time |
| MMAS | Heu | 53538.4 | 5.44% | 1.59s | 276518.6 | 6.75% | 5.60s | 465673.6 | 23.18% | 32.61s |
| | Heu+2opt | 50902.8 | 0.25% | 1.78s | 260587.3 | 0.60% | 4.72s | 381135.2 | 0.82% | 10.55s |
| | Heu+3opt | 50778.0 | 0.00% | 4.01s | 259311.8 | 0.10% | 14.50s | 378927.3 | 0.24% | 23.67s |
| MCTS | MCTS | 50935 | 0.31% | 44.2s | 265784 | 2.60% | 1.67m | 384727 | 1.77% | 3.99m |
| AttGCN | | 71257 | 40.33% | − | 450576 | 73.94% | − | 719161 | 90.24% | − |
| DIMES | | 86730 | 70.80% | − | 559558 | 116.01% | − | 1061612 | 180.83% | − |
| UTSP | NAR+G | 60951 | 20.03% | − | 305828 | 18.06% | − | 459955 | 21.67% | − |
| DIFUSCO | | 64691 | 27.40% | − | 314332 | 21.34% | − | 449435 | 18.89% | − |
| AttGCN | | 52080.3 | 2.56% | +1.13s | 275918.1 | 6.51% | +3.57s | 434553.4 | 14.95% | +16.13s |
| DIMES | | 52412.0 | 3.22% | +1.11s | 338302.8 | 30.60% | +4.05s | 716813.4 | 89.62% | +18.65s |
| UTSP | + HEATACO | 53194.9 | 4.76% | +1.09s | 275230.1 | 6.25% | +3.52s | 444158.5 | 17.49% | +14.99s |
| DIFUSCO | | 51541.7 | 1.50% | +1.16s | 268323.6 | 3.58% | +3.52s | 409132.1 | 8.23% | +15.99s |
| AttGCN | | 50902 | 0.24% | +44.2s | 265338 | 2.43% | +1.67m | 388518 | 2.77% | +3.99m |
| DIMES | | 50856 | 0.25% | +44.2s | 263164 | 1.59% | +1.67m | 386985 | 2.37% | +3.99m |
| UTSP | +MCTS | 51060 | 0.56% | +44.2s | 264061 | 1.94% | +1.67m | 385057 | 1.86% | +3.99m |
| DIFUSCO | | 50908 | 0.26% | +44.2s | 262472 | 1.32% | +1.67m | 387623 | 2.54% | +3.99m |
| AttGCN | | 50935.9 | 0.31% | +1.48s | 261034.6 | 0.77% | +4.51s | 381053.1 | 0.80% | +9.88s |
| DIMES | + HEATACO | 50927.6 | 0.29% | +1.61s | 282104.9 | 8.90% | +5.76s | 551552.4 | 45.90% | +17.84s |
| UTSP | +2opt | **50886.3** | **0.21%** | +1.34s | 260868.9 | 0.70% | +3.81s | **380555.5** | **0.67%** | +8.33s |
| DIFUSCO | | 50931.4 | 0.30% | +1.52s | **260811.5** | **0.68%** | +3.70s | 380956.3 | 0.77% | +8.77s |
| AttGCN | | 50778.7 | 0.0014% | +3.65s | 259131.3 | **0.0333%** | +14.30s | 379060.0 | 0.2719% | +23.58s |
| DIMES | + HEATACO | **50778.0** | **0.0000%** | +3.97s | 270818.6 | 4.5450% | +18.18s | 460909.2 | 21.9233% | +42.77s |
| UTSP | +3opt | **50778.0** | **0.0000%** | +3.95s | 259236.0 | 0.0737% | +14.25s | 378772.9 | **0.1960%** | +22.49s |
| DIFUSCO | | 50778.7 | 0.0014% | +3.74s | 259185.7 | 0.0543% | +13.10s | 379009.1 | 0.2585% | +25.77s |

*Table 6.* Candidate sparsity and auxiliary cross-entropy diagnostics on TSPLIB at $\epsilon_h = 10^{-4}$. Edges/$N$ reports $|E_{\epsilon_h}(H)|/N$ on the *raw* thresholded graph (no per-node cap; lower is sparser). Missing (%) reports $100 \cdot \text{Miss}(H)/N$ (lower is better). CE and WCE report the binary cross entropy and class-weighted cross entropy of the original heatmap against the TSPLIB optimal-tour adjacency (lower indicates better alignment with reference labels). As discussed in the main paper, CE/WCE are only auxiliary, label-based indicators: they can help contextualise heatmap quality when benchmark tours are available, but they do not fully determine decoding performance.

| Method | pcb442 | | | | pr1002 | | | | pr2392 | | | |
|---|---|---|---|---|---|---|---|---|---|---|---|---|
| | Edges/$N$ | Missing | CE | WCE | Edges/$N$ | Missing | CE | WCE | Edges/$N$ | Missing | CE | WCE |
| AttGCN | 4.65 | 0.6787% | **0.008499** | **0.797472** | 4.72 | 2.3952% | 0.0045336 | 0.99029 | 4.77 | 1.7140% | **0.00176** | **0.90953** |
| DIMES | 3.75 | 2.2624% | 0.009298 | 0.911116 | 116.34 | 7.5848% | 0.0072196 | 1.66238 | 493.26 | 16.3043% | 0.00476 | 2.66811 |
| UTSP | 40.68 | 0.0000% | 0.014476 | 1.36388 | 69.00 | 0.0000% | 0.0076514 | 1.69276 | 112.63 | 0.0000% | 0.0043 | 2.47135 |
| DIFUSCO | 3.83 | 1.5837% | 0.010199 | 1.02956 | 9.56 | 0.3992% | **0.0041682** | **0.92636** | 23.43 | 0.0000% | 0.00193 | 1.00679 |

into the lowest-confidence bins (Figure 8), increasing the number of competing low-signal edges that the decoder must disambiguate. In contrast, AttGCN and DIFUSCO remain comparatively stable in sparsity, but still show a visible degradation in the low-confidence tail structure on pr1002/pr2392, consistent with the weaker decoding robustness of greedy merge (Table 5).

The failure case for DIMES on TSPLIB (Table 5) aligns with these diagnostics: severe candidate densification and nontrivial optimal-tour miss rates reduce the effectiveness of both sampling and local improvement restricted to the candidate graph.

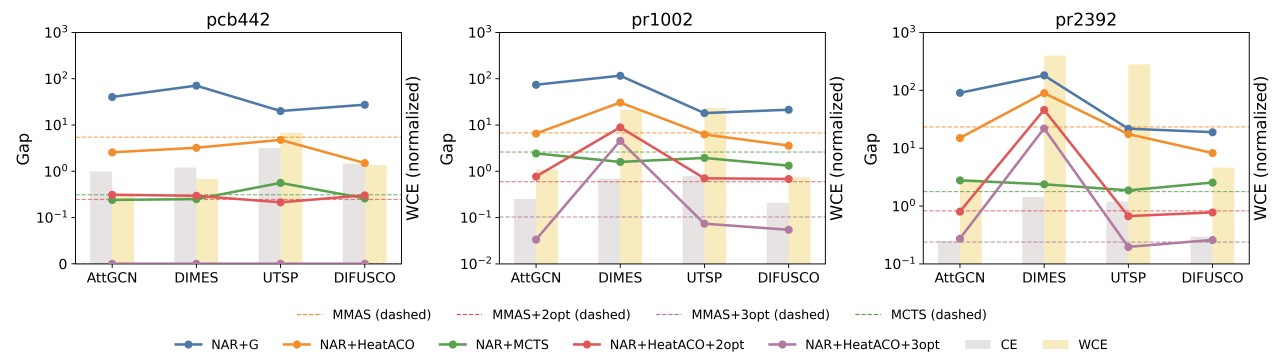

*Figure 7.* Decoding gaps on TSPLIB versus auxiliary CE/WCE diagnostics (Table 6). Lines show the optimality gap (%) achieved by different decoders for each heatmap source; bars show CE and WCE (normalised per instance).

*(a)* AttGCN

*(b)* DIMES

*(c)* UTSP

*(d)* DIFUSCO

*Figure 8.* Interval contribution analysis on TSPLIB instances (pcb442, pr1002, pr2392). Each row corresponds to a heatmap predictor and shows how candidate edges distribute across confidence intervals (bars) and how optimal-tour edges concentrate (curve; normalised over optimal-tour edges).

## F. Convergence and Local Improvement

**Connection to the main paper.**    This section extends the time-to-quality results by adding convergence curves when local improvement is enabled.

All convergence plots use the y-axis definition described in Figure 2. The main paper reports convergence without local improvement across scales; Figure 9 reports the corresponding curves when applying a lightweight 2-opt pass at each iteration. With local improvement enabled, absolute gaps shrink and the curves become more tightly clustered, but the ranking is largely preserved: HEATACO retains a consistent time-to-quality advantage because the heatmap prior steers sampling toward high-recall regions where 2-opt can be effective early. Notably, the gaps between methods persist well into the mid/late iterations, indicating that pheromone feedback continues to matter even when a computationally inexpensive local search is applied—it helps consolidate globally consistent edges rather than relying solely on local geometric repairs.

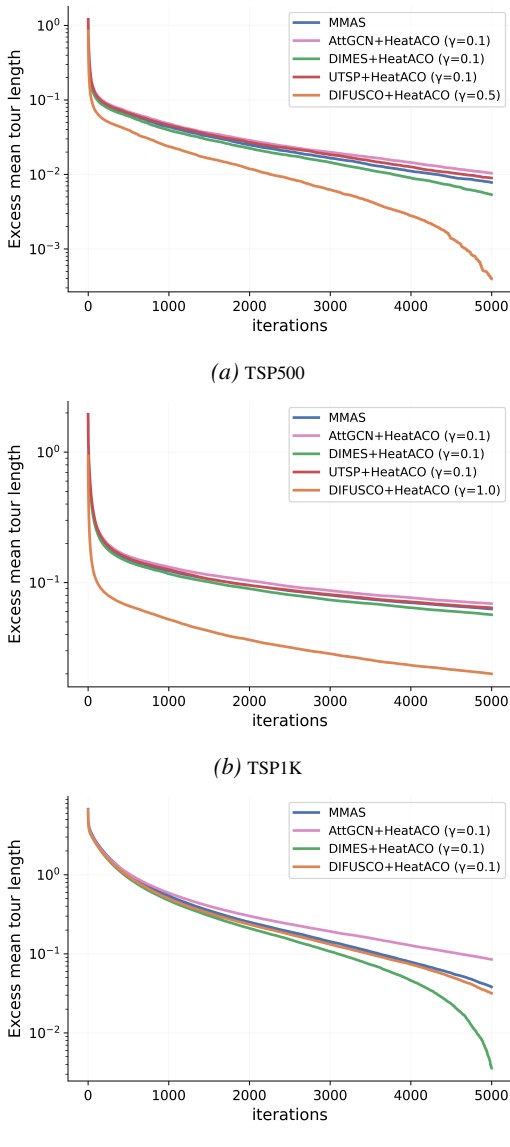

*(a)* TSP500

*(b)* TSP1K

*(c)* TSP10K

*Figure 9.* Convergence of MMAS and HEATACO with 2-opt local improvement across scales.

# G. Additional Ablations and Sensitivity

**Connection to the main paper.** This section complements the ablation and sensitivity results in Section 6 with additional sweeps and local-search variants.

**Component ablations.** The main paper reports the component ablation study (Table 2), quantifying the roles of the heatmap prior ($\gamma$) and the distance heuristic ($\beta$) in Eq. (9).

## G.1. Coarse Tuning of the Heatmap Exponent $\gamma$

The exponent $\gamma$ in Eq. (9) controls how strongly the heatmap biases sampling. Different predictors can yield heatmaps with different calibration/sharpness, so $\gamma$ may need mild adjustment. We find that a *coarse* sweep over $\gamma \in \{0.1, 0.5, 1.0, 2.0\}$ is sufficient to obtain strong results. The main paper reports the no-local-search sensitivity across scales in Figure 4.

**No local search: scale and miscalibration.** Without local improvement, $\gamma$ directly controls the exploration–exploitation trade-off in sampling. On large instances, stronger guidance can help underconfident heatmaps by reducing the effective branching factor and accelerating pheromone learning; for example, on TSP10K, AttGCN improves from 28.1% gap at $\gamma=0.1$ to 19.3% at $\gamma=2.0$, and DIMES improves from 25.2% at $\gamma=0.1$ to 8.8% at $\gamma=1.0$. However, overly large $\gamma$ can over-amplify spuriously confident wrong edges and reduce the chance that pheromone feedback corrects early mistakes; for instance, on TSP1K with DIMES, the gap increases from 5.2% at $\gamma=0.1$ to 13.6% at $\gamma=0.5$. Moreover, when the prior is systematically weak (e.g., DIFUSCO on TSP10K), tuning $\gamma$ alone yields limited gains without local improvement, motivating the 2-opt/3-opt results in the main paper.

**Interaction with local search.** With 2-opt/3-opt enabled, the heatmap mainly serves as a proposal prior over a sparse candidate graph, while local improvement repairs many residual geometric conflicts. In this regime, overly large $\gamma$ can reduce tour diversity and over-commit to spurious high-confidence edges, which is particularly harmful because both sampling and local moves are restricted to the candidate graph. This effect is most pronounced on large instances: on TSP10K, the AttGCN heatmap with HEATACO+2opt degrades from 1.27% at $\gamma=0.1$ to 5.79% at $\gamma=2.0$, and the DIMES heatmap with HEATACO+2opt degrades from 1.15% at $\gamma=0.1$ to 6.73% at $\gamma=1.0$. The same trend appears for 3-opt (e.g., DIMES degrades from 0.40% at $\gamma=0.1$ to 3.50% at $\gamma=1.0$ on TSP10K).

**Source-dependent behavior.** For AttGCN/DIMES, small $\gamma$ (often $\gamma=0.1$) is consistently strong under 2-opt/3-opt across scales. UTSP becomes comparatively insensitive once local search is enabled (gaps vary only slightly across $\gamma$), while some heatmaps can prefer moderate $\gamma$ on smaller instances (e.g., DIFUSCO on TSP1K).

## G.2. Entropy-targeted selection of $\gamma$ (label-free)

The sensitivity results above suggest that $\gamma$ should leverage the heatmap ranking while avoiding an overly sharp proposal distribution (especially with local improvement), which can reduce tour diversity and over-commit to spuriously confident edges. We interpret $\gamma$ as a temperature in log space: for $H_{ij} \in (0, 1]$, $H_{ij}^\gamma = \exp(\gamma \log H_{ij})$, so increasing $\gamma$ sharpens the heatmap-induced distribution.

We propose a *label-free* rule that chooses $\gamma$ by targeting the entropy (effective support size) of a heatmap-only proposal distribution. For each node $i$, let $\mathcal{N}_i^H \subseteq C_i$ denote the heatmap-derived portion of the decoder candidate list (Appendix B); we ignore distance-based fallback candidates to isolate the effect of heatmap sharpening. We normalise scores within $\mathcal{N}_i^H$ as

$$\tilde{h}_{ij} \triangleq \frac{H_{ij}}{\max_{k \in \mathcal{N}_i^H} H_{ik} + \delta}, \quad j \in \mathcal{N}_i^H, \qquad (15)$$

where $\delta$ is a small constant (we use $\delta = 10^{-12}$). We then define a heatmap-only proposal distribution

$$q_{ij}(\gamma) \triangleq \frac{\tilde{h}_{ij}^\gamma}{\sum_{k \in \mathcal{N}_i^H} \tilde{h}_{ik}^\gamma}, \quad j \in \mathcal{N}_i^H, \qquad (16)$$

and its effective support size

$$S_i(\gamma) \triangleq \exp\left(-\sum_{j \in \mathcal{N}_i^H} q_{ij}(\gamma) \log q_{ij}(\gamma)\right). \qquad (17)$$

Intuitively, $S_i(\gamma)$ measures the number of candidates that meaningfully contribute under node $i$. We aggregate by $S(\gamma) \triangleq \mathrm{median}_i\, S_i(\gamma)$ (over nodes with $|\mathcal{N}_i^H| \geq 2$) and choose $\gamma$ such that $S(\gamma) \approx S^*$. Since $S(\gamma)$ decreases as $\gamma$ increases, $\gamma$ can be selected either by bisection (if treated as continuous) or, in our setting, by evaluating the same coarse grid $\gamma \in \{0.1, 0.5, 1, 2\}$ and taking the closest value.

**Recommended defaults.** Based on our sensitivity observations, we use different entropy targets depending on whether local improvement is enabled: with 2-opt/3-opt, we recommend $S_{\mathrm{LS}}^* \in [6, 10]$ (default $S_{\mathrm{LS}}^* = 8$) to preserve proposal diversity; without local search, we recommend $S_{\mathrm{noLS}}^* \in [3, 6]$ (default $S_{\mathrm{noLS}}^* = 4$) to allow stronger guidance and faster pheromone learning on large instances.

# H. Machine Learning for TSP

Machine learning for the Travelling Salesman Problem (TSP) is a central instance of learning-based combinato-

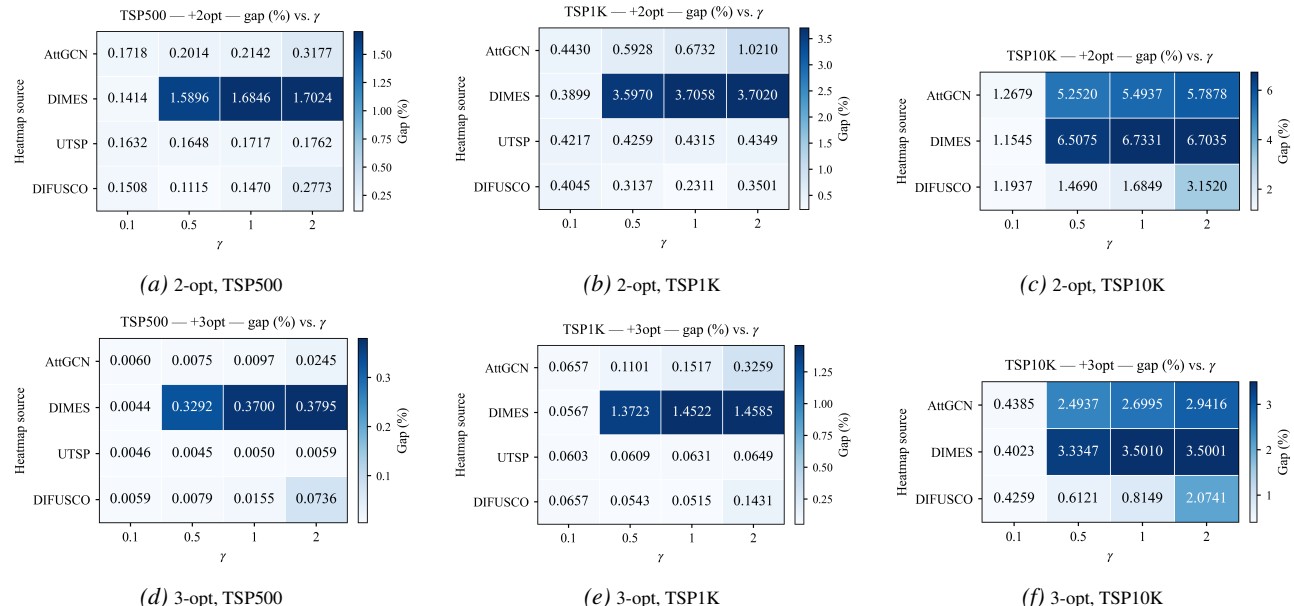

*Figure 10.* Sensitivity of HEATACO with local improvement to $\gamma$ across instance scales (gap %). UTSP does not provide TSP10K heatmaps. Top row: 2-opt. Bottom row: 3-opt.

rial optimisation; see the methodological overview by Bengio et al. (Bengio et al., 2021) and the NCO survey of Liu et al. (Liu et al., 2023). Despite rapid progress, classical solvers such as Concorde (Applegate et al., 2006) and LKH-3 (Helsgaun, 2017) remain the dominant references for Euclidean TSP, so many ML systems are best understood as learning reusable priors that reduce the search effort required to approach these baselines under realistic CPU budgets.

Early neural approaches constructed tours autoregressively, from Pointer Networks (Vinyals et al., 2015) to reinforcement-learning based solvers (Bello et al., 2017) and policy-gradient learning of heuristics (Deudon et al., 2018). This line was strengthened by attention/transformer architectures (Kool et al., 2019; Bresson & Laurent, 2021) and hybrid guided search schemes that combine learned scores with tree search (Li et al., 2018). Subsequent work has focused on improving efficiency and generalisation across distributions and scales, for example by exploiting symmetry (Kim et al., 2022), emphasising sparsification for scalable attention and GNNs (Lischka et al., 2024), rescaling via subgraph mechanisms (Huang et al., 2025), and building unified models that transfer across instance sizes (Xiao et al., 2025). Beyond one-shot construction, many methods explicitly learn or guide improvement steps, including learning local rewriting policies (Chen & Tian, 2019), learning 2-opt style improvement operators (Costa et al., 2020), GNN-guided local search (Hudson et al., 2022), and efficient active search/test-time refinement for combinatorial optimization (Hottung et al., 2022).

Large-scale TSP further motivates hierarchical and divide-and-conquer strategies that reduce the effective problem size, e.g., hierarchical RL construction (Pan et al., 2023), select-and-optimise pipelines (Cheng et al., 2023), unified neural divide-and-conquer frameworks (Zheng et al., 2025), and dual divide-and-optimise algorithms (Zhou et al., 2025b). Related trends include unifying multiple CO problems through reduction to a general TSP representation (Pan et al., 2024) and training broader routing models (e.g., foundation-model style solvers and edge-based transformers) for TSP/VRP families (Berto et al., 2024; Meng et al., 2025).

In parallel, non-autoregressive pipelines predict global structure in one forward pass (e.g., dense edge scores) and defer feasibility to a decoder (Joshi et al., 2019). This prediction–decoding separation is especially prominent in large-scale heatmap paradigms, where predictors such as AttGCN, DIMES, UTSP, and diffusion-based models like DIFUSCO produce edge-confidence matrices that are then converted into tours (Fu et al., 2021; Qiu et al., 2022; Min et al., 2023; Sun & Yang, 2023; Wang et al., 2025). Once a heatmap is fixed, decoding and test-time search become first-class algorithmic components: greedy merging is computationally efficient but can be fragile, while local-improvement decoders use the heatmap to bias proposals; AttGCN popularised MCTS-guided $k$-opt decoding (Fu et al., 2021). However, MCTS pipelines introduce substantial design and tuning choices (Browne et al., 2012), and recent analyses stress that search configuration can dominate performance and should be standardized for fair heatmap comparisons (Xia et al., 2024; Pan et al., 2025). More broadly,

learning-to-search frameworks include learning combinatorial optimisation algorithms over graphs (Khalil et al., 2017), efficiency-oriented abstractions such as bisimulation quotienting (Drakulic et al., 2023), learnable meta-optimizers (Dernedde et al., 2024), and probabilistic generative approaches such as GFlowNets (Zhang et al., 2023), including ACO-style sampling coupled with GFlowNet training (Kim et al., 2024).

Our contribution is orthogonal to predictor training: HEAT-ACO revisits the heatmap-to-tour stage and proposes an ACO/MMAS-based decoder that treats the heatmap as a soft prior while using pheromone feedback as lightweight, instance-specific global conflict resolution. This framing connects to hybrid search methods and ACO-inspired neural combinatorial optimization (Kim et al., 2024), but targets a plug-and-play decoder that can be applied across predictors and instance distributions.

