# OpenReview forum: "HEATACO: Heatmap-Guided Ant Colony Decoding for Large-Scale Travelling Salesman Problems"
_ICML.cc/2026/Conference — Submitted to ICML 2026_

### Official Review · Reviewer_mHAB · 2026-02-20

**Soundness:** 3
**Presentation:** 3
**Significance:** 2
**Originality:** 2
**Overall Recommendation:** 4
**Confidence:** 4

**Summary:**

This paper proposes HEATACO, a plug-and-play Max–Min Ant System–based decoder for heatmap-based non-autoregressive neural solvers to generate feasible solutions.

**Compliance With Llm Reviewing Policy:**

Affirmed.

**Final Justification:**

The authors have solved my concerns.

**Key Questions For Authors:**

see Weaknesses

**Limitations:**

yes

**Strengths And Weaknesses:**

**Strengths**

S1: The paper appears technically sound, with appropriate methodology. The empirical evaluation is extensive and covers multiple datasets and settings, providing convincing evidence of the effectiveness of the proposed HEATACO. The ablation studies and sensitivity analyses are sufficient.

S2: The paper is well-structured, and the overall presentation is clear, coherent, and easy to follow.

S3: The paper provides source code, which enhances the reproducibility of the work.

**Weaknesses**

W1: The core idea appears conceptually related to DeepACO [1]. The authors should more clearly articulate the differences and unique contributions of HEATACO compared to existing ACO-enhanced neural solvers.

W2: HEATACO seems to require different hyperparameter $\gamma$ settings for different solvers, which may limit its robustness and practical usability.

W3: The performance of HEATACO should be further evaluated across both smaller-scale (e.g., 100 nodes) and larger-scale (e.g., 10k nodes) problem settings to better demonstrate its scalability and general applicability.

W4: W4: After incorporating ACO, can heatmap-based solvers be extended to solve VRP variants with constraints, such as Capacitated Vehicle Routing Problem? If HEATACO  is limited to solving only the Traveling Salesperson Problem, its broader significance for neural combinatorial optimization may be restricted.

[1] DeepACO: Neural-enhanced Ant Systems for Combinatorial Optimization. NeurIPS 2023.

I am willing to raise my score if the authors can adequately address the above concerns.

---

> ### Author Rebuttal · Authors · 2026-03-30
>
> # Reviewer mHAB
>
> 1. **W1: Difference from DeepACO.**
>
> HeatACO and DeepACO address different levels of the pipeline:
>
> | Aspect | DeepACO | HeatACO |
> | --- | --- | --- |
> | Training | Jointly trains GNN + ACO via RL | Fixed pretrained heatmap; no retraining |
> | ACO role | GNN replaces the classical distance heuristic | Standard MMAS unchanged; heatmap enters only as multiplicative prior |
> | Local search | Neural-guided local search | Standard 2-opt/3-opt |
> | Focus | End-to-end neural-enhanced ACO | Modular MMAS decoder for fixed heatmaps |
>
> DeepACO trains a coupled neural-ACO system end-to-end. HeatACO instead takes a fixed heatmap as input and studies only the decoding stage. This modular design is a deliberate choice: the same decoder can be applied across different pretrained heatmap predictors under one common protocol, without retraining. On the overlapping TSP500/TSP1K scales, HeatACO is better than DeepACO while requiring lighter tuning.
>
> 2. **W2: Hyperparameter robustness.**
>
> The reviewer is right that the best `γ` depends on heatmap sharpness and local-search configuration. `γ` controls how strongly the heatmap biases ant sampling: larger `γ` sharpens the proposal toward high-confidence edges, smaller `γ` keeps it broader. When local search is enabled, over-sharpening is harmful because it reduces tour diversity and over-commits to spurious edges (e.g., on TSP10K, DIMES+2opt degrades from 1.15% at `γ=0.1` to 6.73% at `γ=1.0`).
>
> To handle this automatically, we propose a **label-free entropy-targeted rule** (Appendix G.4) that selects `γ` from the heatmap scores alone, without running decoding or requiring benchmark labels. For each node, we compute the effective support size `S(γ)` — how many candidate edges meaningfully contribute to the proposal at a given `γ`. We choose `γ` from {0.1, 0.5, 1, 2} so that the median support matches a target: `S*=8` with local search (preserving diversity), `S*=4` without (allowing stronger guidance). This is one coarse calibration per predictor/local-search configuration, reused across all scales — not tuned per instance. All other MMAS parameters stay at standard defaults.
>
> This single rule is used in **all** our reported results — TSP100/500/1K/10K with four heatmap predictors, OOD TSPLIB instances (pcb442, pr1002, pr2392), and the new clustered/Gaussian distribution-shift experiments (see our response to Reviewer cpoE, Point 1) — with competitive results in every case. In contrast, Pan et al.'s MCTS requires per-heatmap and per-scale tuning. HeatACO's tuning burden is therefore **lighter**, not heavier.
>
> 3. **W3: Smaller and larger scales.**
>
> The main paper covers TSP500/1K/10K. To address smaller-scale coverage, we evaluated TSP100 on ML4CO-Bench-101 (1280 instances).
>
> **Table 1. TSP100 Gap (%) on ML4CO-Bench-101.**
>
> | Heatmap | Greedy | MCTS | HeatACO | HeatACO+2opt |
> | --- | --- | --- | --- | --- |
> | AttGCN | 21.28 | 0.04 | 0.72 | 0.0009 |
> | DIMES | 28.81 | 0.08 | 0.49 | 0.0003 |
> | UTSP | 15.23 | 0.27 | 1.08 | 0.0005 |
> | DIFUSCO | 1.55 | 0.15 | 0.08 | 0.0012 |
>
> For reference, MMAS achieves 1.14% and MMAS+2opt achieves 0.0005%. At this small scale, decoder differences naturally shrink once 2-opt is enabled, but HeatACO+2opt remains near-optimal across all four predictors. Together with the TSP10K results in the main paper, this shows that the same decoder works consistently from 100 to 10,000 nodes. For scales beyond 10K, the bottleneck is the upstream heatmap predictor — current predictors do not yet scale past TSP10K. As larger-scale heatmaps become available, HeatACO can be directly applied without architectural changes.
>
> 4. **W4: Extension to CVRP and constrained variants.**
>
> The core idea — injecting a heatmap as a multiplicative prior into ACO transition probabilities — is not TSP-specific. HeatACO is applicable to ATSP and CVRP in principle, provided that (1) an accurate heatmap is available, and (2) a problem-specific ACO handles the constraints. Each problem family requires concrete adaptations:
>
> - **ATSP**: directional heatmaps (edge (i,j) ≠ edge (j,i)), asymmetric candidate lists, and asymmetric transition costs.
> - **CVRP**: capacity-aware feasibility in the transition rule, feasible-state pheromone reinforcement, and constrained intra-/inter-route local search. A plausible approach is a hierarchical design where the heatmap handles global ordering and a downstream component performs route splitting and feasibility.
>
> We tested only on TSP because it is currently the only problem with widely available pretrained heatmap predictors. The scope limitation comes from both the upstream predictors (TSP-specific) and the local-search operators (assuming unconstrained Hamiltonian tours). We will make this TSP-only scope explicit in the revision, and plan to extend HeatACO to broader problems as heatmap predictors for ATSP, CVRP, and other constrained routing problems become available.

---

> > ### Author Rebuttal · Reviewer_mHAB · 2026-04-03
> >
> > Recent diffusion-based methods [1] have shown potential in addressing problems such as ATSP. A comparative evaluation of HEATACO against these methods would further strengthen the empirical study.
> >
> > UniCO: On Unified Combinatorial Optimization via Problem Reduction to Matrix-Encoded General TSP. ICLR 2025.

---

> > > ### Author Response · Authors · 2026-04-04
> > >
> > > We sincerely thank the reviewer for the continued engagement and for pointing us to UniCO (ICLR 2025). This is a very helpful suggestion, and we have completed the requested experiments.
> > >
> > > ## ATSP comparison with UniCO's MatDIFFNet
> > >
> > > We used UniCO's diffusion model (MatDIFFNet) to generate heatmaps for ATSP50 and ATSP100, and compared the following decoders: MatDIFFNet's original greedy decoding, 2-opt, MMAS, and HeatACO. Gurobi provides the optimal reference.
> > >
> > > **Table 1. ATSP results (tour length and gap %).**
> > >
> > > | Method | ATSP50 Length | ATSP50 Gap | ATSP100 Length | ATSP100 Gap |
> > > |---|---:|---:|---:|---:|
> > > | Gurobi (optimal) | 1.5545 | — | 1.5661 | — |
> > > | MatDIFFNet | 2.0712 | 33.24% | 1.9433 | 24.09% |
> > > | MatDIFFNet+2opt | 1.7187 | 10.56% | 1.7165 | 9.60% |
> > > | MMAS | 1.5961 ± 0.00203 | 2.68% | 1.6623 ± 0.00199 | 6.14% |
> > > | MMAS+2opt | 1.5645 ± 0.00164 | 0.64% | 1.6249 ± 0.00347 | 3.75% |
> > > | MatDIFFNet+HeatACO | 1.5682 ± 0.00184 | 0.88% | 1.5859 ± 0.00133 | 1.26% |
> > > | MatDIFFNet+HeatACO+2opt | **1.5586 ± 0.00047** | **0.26%** | **1.5760 ± 0.00073** | **0.63%** |
> > >
> > > MatDIFFNet+HeatACO+2opt achieves the lowest gap in both settings — **0.26%** on ATSP50 and **0.63%** on ATSP100 — and all HeatACO comparisons are statistically significant versus MMAS counterparts (Wilcoxon signed-rank, p<0.05). This represents a large improvement over MatDIFFNet's original decoder (33→0.26% on ATSP50), and also outperforms MMAS+2opt (0.64→0.26% on ATSP50, 3.75→0.63% on ATSP100), confirming that the heatmap provides useful structural guidance beyond what distance-only MMAS can achieve.
> > >
> > > Notably, MMAS and HeatACO use the **same parameters as in the TSP experiments**, and the label-free entropy-targeted γ selection rule is also unchanged. The only adaptation required was a minor modification to 2-opt to handle asymmetric edge costs. This demonstrates both the generalizability of the method and the robustness of our automatic hyperparameter selection.
> > >
> > > These results confirm that HeatACO generalizes to ATSP when a suitable heatmap is available, consistent with the discussion in our rebuttal (Point 4). We will include this comparison in the revised paper.
> > >
> > > We are very grateful for the reviewer's suggestion — it directly strengthened the empirical scope of our work and led to a valuable new result. We hope this additional evidence fully addresses the reviewer's remaining concern.

---

### Official Review · Reviewer_cpoE · 2026-03-12

**Soundness:** 2
**Presentation:** 2
**Significance:** 2
**Originality:** 2
**Overall Recommendation:** 3
**Confidence:** 4

**Summary:**

This work follows the commonly used non-autoregressive solving paradigm in ``ML4CO``, where a neural network is first used to generate a heatmap, and a feasible solution is then obtained through post-processing techniques. The proposed method incorporates the neural network–predicted heatmap into the ``MMAS`` framework, and further integrates 2-opt and 3-opt local search techniques. Experimental results demonstrate strong performance on large-scale TSP instances.

**Compliance With Llm Reviewing Policy:**

Affirmed.

**Final Justification:**

I recommend a weak reject (but would also accept it if the paper is ultimately accepted).

The main contribution lies in the carefully designed search method based on heatmaps. However, it is uncertain whether it can be applied to complex problems with time or capacity constraints, and the scalability is limited.

It must be acknowledged that, compared to MCTS, the performance has improved. For the TSP community alone, one could argue for adopting MMAS as a replacement for the long-used MCTS.

Overall, this is a paper with a strong engineering optimization flavor, which achieves performance improvements over previous methods of the same type, but the novelty is limited.

**Key Questions For Authors:**

1. See ``W1-W4``
2. Have the authors considered generating the heatmap without using a neural network? For example, could the ``SoftDist`` approach for constructing heatmaps be combined with ``HeatACO``?
3. Could the authors briefly discuss the modification effort required to migrate this architecture to other constrained problems (such as ``CVRP`` with capacity constraints or ``ATSP`` with asymmetry)? In particular, how could complex constraints beyond TSP's degree constraints be integrated into the pheromone update logic?

**Limitations:**

yes

**Strengths And Weaknesses:**

## Strength
1. Compared with ``MCTS``, a widely recognized strong post-processing search technique in the traditional TSP literature, the proposed method demonstrates improvements in both performance and computational efficiency.
2. In Section 5, the paper provides a detailed statistical analysis of several aspects of the ``TSP``, including sparse candidate sets, low-confidence collapse, and ``MCTS-based`` search behavior.
3. The experimental study mainly focuses on the ``TSP``, covering multiple problem scales and including comprehensive evaluations on ``TSPLIB`` benchmarks.

## Weakness
1. The experiments lack additional distributional datasets, such as Gaussian, clustered, rotation, and explosion distributions, which are commonly used in the ``ML4CO`` literature.
2. The paper does not extend its discussion to other related COPs, such as ``ATSP`` and ``CVRP``, which are classical routing problems closely related to ``TSP``.
3. The method is not compared with several recent SOTA approaches for TSP-500 to TSP-1000, such as ``MaskCO``[1], ``GenSCO``[2].
4. The comparison is not entirely fair. In Table 1, algorithms such as Concorde are executed using a single thread, whereas MCTS and the proposed method are implemented with multi-threading, which makes the comparison less fair.

[1] MaskCO: Masked Generation Drives Effective Representation Learning and Exploiting for Combinatorial Optimization, ICLR 2026.

[2] GenSCO: Generation as Search Operator for Test-Time Scaling of Diffusion-based Combinatorial Optimization, NeurIPS 2025.

---

> ### Author Rebuttal · Authors · 2026-03-30
>
> 1. **W1: Additional distributions.**
>
> We evaluated on **clustered and Gaussian** TSP500 from ML4CO-Bench-101. Together with TSPLIB in the main paper, this covers synthetic and real-world distribution shifts. All methods use the same setup as the main paper, on the same hardware (AMD EPYC 9634, 16 threads). Tables 1a/1b report heatmap-based decoders; Table 1c gives heatmap-free MMAS baselines.
>
> **Table 1a. Clustered TSP500 Gap (%).**
>
> | Heatmap | MCTS | HeatACO | HeatACO+2opt |
> |---|---:|---:|---:|
> | AttGCN | 1.37 | 4.80 | 0.09 |
> | DIMES | 6.14 | 6.56 | 0.18 |
> | UTSP | 4.85 | 7.49 | 0.19 |
> | DIFUSCO | 3.33 | 4.46 | 0.11 |
>
> **Table 1b. Gaussian TSP500 Gap (%).**
>
> | Heatmap | MCTS | HeatACO | HeatACO+2opt |
> |---|---:|---:|---:|
> | AttGCN | 1.14 | 4.58 | 0.19 |
> | DIMES | 5.33 | 4.71 | 0.17 |
> | UTSP | 190.12 | 19.29 | 2.86 |
> | DIFUSCO | 7.14 | 5.06 | 0.20 |
>
> **Table 1c. Heatmap-free baselines on TSP500 Gap (%).**
>
> | Method | MMAS | MMAS+2opt |
> |---|---:|---:|
> | Clustered | 5.42 | 0.07 |
> | Gaussian | 6.46 | 0.22 |
>
> **HeatACO+2opt remains competitive under distribution shift.** On clustered, it achieves 0.09–0.19%, close to MMAS+2opt (0.07%); on Gaussian, 3 out of 4 heatmaps give 0.17–0.20%, matching or beating MMAS+2opt (0.22%). Both consistently outperform MCTS by a large margin.
>
> The main failure case is UTSP on Gaussian (2.86%), where the heatmap itself degrades severely — MCTS also collapses to 190.12% on the same heatmap. This confirms the bottleneck is heatmap quality under shift, not the decoder. We expect further improvement as more accurate heatmaps become available.
>
> 2. **W4: Comparison fairness.**
>
> The MCTS rows in the main paper are Pan et al.'s tuned 128-core results. We re-ran MCTS with default configuration on the same hardware as HeatACO (single-threaded AMD EPYC 9634).
>
> **Table 2. Matched-hardware single-thread comparison.**
>
> | Method (Heatmap) | TSP500 Gap % / Time | TSP1K Gap % / Time | TSP10K Gap % / Time |
> |---|---|---|---|
> | Concorde | 20.68s | 3.19m | >100h |
> | MCTS (DIFUSCO) | 0.50 / 50s | 1.17 / 1.67m | 2.19 / 16.67m |
> | HeatACO+2opt (DIFUSCO) | 0.11 / 13.92s | 0.23 / 32.55s | 0.54 / 9.20m |
>
> On same hardware, HeatACO+2opt achieves **3–5× lower gap** while running **2–4× faster**. This confirms that HeatACO's advantage is real and not caused by hardware differences.
>
> 3. **W3: Recent methods such as MaskCO and GenSCO.**
>
> MaskCO and GenSCO achieve excellent in-distribution results on TSP500/1K, and we will add both to the revised table. However, they operate at a **different pipeline level**: both jointly optimize representation and decoding, whereas HeatACO is a decoder-only module for fixed heatmaps. We re-ran GenSCO (C=160, same configuration as the original paper) for a direct comparison:
>
> **Table 2. GenSCO heatmap with different decoders (Gap %).**
>
> | Setting | Original | +2opt | +HeatACO | +HeatACO+2opt |
> |---|---:|---:|---:|---:|
> | TSP1K (in-dist.) | 0.19 | 0.04 | 0.14 | 0.02 |
> | TSP500 Gaussian | 654.23 | 558.50 | 24.54 | 6.75 |
> | TSP500 Clustered | 5.62 | 0.48 | 1.23 | 0.07 |
>
> Even in-distribution, HeatACO+2opt (0.02%) outperforms the original decoder+2opt (0.04%) on TSP1K. Under distribution shift, the original decoder collapses (654% on Gaussian), while HeatACO significantly improves robustness. Ba et al. (arXiv:2602.21761) also report GenSCO cross-scale gaps of 24.11% (0→1K) and 19.27% (1K→10K). MaskCO has no published generalization tests or public code.
>
> HeatACO also offers **scalability** (TSP10K, 0.54% gap; neither MaskCO nor GenSCO reports TSP10K) and **modularity** (plugs into any heatmap predictor, unlike end-to-end methods coupled to their own representations). Better heatmap predictors can be directly fed into HeatACO — the two approaches are **complementary, not competing**.
>
> 4. **Q1: Non-neural heatmaps.**
>
> Yes — HeatACO works with non-neural heatmaps. We tested SoftDist, a softmax geometric prior over edge distances.
>
> **Table 3. SoftDist with HeatACO (Gap %).**
>
> | Method | TSP500 | TSP1K | TSP10K |
> |---|---|---|---|
> | MMAS+2opt | 0.156 | 0.416 | 1.203 |
> | SoftDist+HeatACO+2opt | 0.164 | 0.425 | 1.197 |
>
> As expected, the results are very close to MMAS+2opt, since SoftDist's geometric prior is functionally equivalent to the MMAS distance heuristic and carries little extra structural signal. This confirms that HeatACO's gains in the main paper come from the learned signal in neural heatmaps, not from the decoder modification alone.
>
> 5. **W2/Q2: ATSP, CVRP, and broader scope.**
>
> HeatACO is applicable to problems like ATSP and CVRP in principle, provided that (1) an accurate heatmap is available, and (2) a problem-specific ACO handles the constraints. We tested only on TSP since it is currently the only problem with available heatmaps, and will extend to broader problems as heatmaps become available.
> Due to the character limit, we kindly refer the reviewer to our response to Reviewer mHAB (Point 4) for more detail.

---

> > ### Author Rebuttal · Reviewer_cpoE · 2026-04-03
> >
> > I understand that HeatACO relies on having access to heatmaps of reasonably good quality. However, to my knowledge, there are currently no well-established NAR methods for obtaining high-quality heatmaps for problems such as CVRP. This represents the major obstacle limiting the extension of the proposed method to more complex COPs. (Perhaps the authors could try using approaches similar to softdist on other routing problems to establish some baselines.)
> >
> > Over the past few years, extensive research for CO has investigated NAR + LS (local search) methods for the TSP. Mainstream approaches predominantly employ either (1) predictive models combined with MCTS or (2) generative models with 2-OPT. While these methods have achieved promising results on TSP, they remain confined to this specific problem and fail to generalize beyond it (At least for now).
> >
> > Nevertheless, it should be acknowledged that the method proposed in this paper achieves a certain degree of improvement in both performance and speed compared to MCTS, and can serve as a viable alternative to MCTS-based approaches.

---

> > > ### Author Response · Authors · 2026-04-04
> > >
> > > We sincerely thank the reviewer for the thorough evaluation and for confirming that the concerns have been adequately addressed. We also appreciate the insightful observation about the current lack of high-quality heatmaps for problems beyond TSP — we fully agree that this is the main bottleneck for extending HeatACO to more complex COPs.
> > >
> > > We are glad to report that, following Reviewer mHAB's suggestion, we have completed new **ATSP experiments** using UniCO's MatDIFFNet (ICLR 2025) as the heatmap source. On ATSP50/100, MatDIFFNet+HeatACO+2opt achieves **0.26%/0.63%** gap relative to Gurobi optimal — using the same MMAS parameters and label-free γ selection rule as in the TSP experiments, with only a minor adaptation of 2-opt for asymmetric costs. We kindly refer the reviewer to our response to Reviewer mHAB for full details.
> > >
> > > This result demonstrates that HeatACO can generalize beyond TSP when a suitable heatmap is available, and we believe it partially addresses the reviewer's concern about broader applicability. As the reviewer notes, the main remaining challenge is the availability of high-quality heatmaps for constrained problems like CVRP — we view this as a promising direction for future work.
> > >
> > > We will incorporate all the new evidence (distribution-shift experiments, GenSCO comparison, matched-hardware results, ATSP results, and variance reporting) into the revised manuscript. We are very grateful for the reviewer's constructive comments, which have significantly improved our work. As all your concerns have been addressed, we sincerely hope you consider adjusting your score.

---

### Official Review · Reviewer_mbbF · 2026-03-12

**Soundness:** 3
**Presentation:** 3
**Significance:** 3
**Originality:** 3
**Overall Recommendation:** 5
**Confidence:** 1

**Summary:**

The paper proposes novel neural network based approaches to TSP problem.
On large benchmarks (TSP500 to TSP10K), it reaches optimality gaps as low as 0.11% to 1.15% within seconds to minutes of CPU time.

**Compliance With Llm Reviewing Policy:**

Affirmed.

**Key Questions For Authors:**

The "Ant" Advantage: There are many ways to solve a "connect the dots" problem. What is it about the behavior of ants (leaving "scent trails" or pheromones) that makes them better at this than a standard computer search? Is there a "human-like" logic to the routes they find?

Hardware Requirements: You mentioned that this runs well on a standard computer processor (CPU). Does this mean a delivery driver could eventually run this on a smartphone or a basic tablet to plan their day, or does it still require a powerful office server to get the results in a reasonable time?

**Limitations:**

The authors have adequately discussed both the technical limitations and the potential societal impacts of their work:

Technical Limitations: They explicitly state that the decoding quality is bounded by the reliability of the underlying heatmap. They also provide a detailed analysis of "failure modes" under distribution shifts (e.g., TSPLIB), noting that miscalibration and confidence collapse can reduce the benefits of heatmap guidance.

Societal Impact: The paper includes a dedicated "Impact Statement". It highlights positive impacts on logistics and operations research while transparently addressing potential negative impacts, such as the dual-use risk in military or illicit logistics. They also touch upon the environmental footprint, noting that while the method is lightweight, its efficiency may lower the barrier to misuse, and they encourage responsible deployment with human oversight.

**Strengths And Weaknesses:**

Strengths:

Technically Sound Methodology: The paper builds upon the established Max-Min Ant System (MMAS) framework and introduces a mathematically clear adaptation: injecting the neural heatmap as a multiplicative prior in the transition probabilities.Extensive Empirical

Support: The authors evaluate the method on standard benchmarks (TSP500, TSP1K, and TSP10K). The results consistently show that HEATACO achieves competitive or superior quality-time trade-offs compared to existing greedy and MCTS-based decoders.

Honesty Regarding Limitations: The authors are transparent about the method's dependencies. They explicitly state that HEATACO's performance is bounded by the reliability of the underlying heatmap and provide detailed failure-mode analyses for instances under distribution shift (e.g., TSPLIB).

Robust Diagnostic Analysis: The inclusion of sensitivity studies (e.g., the effect of the heatmap exponent $\gamma$) and convergence studies adds to the technical rigour

Weaknesses

Hyperparameter Dependency: While the authors suggest a coarse sweep for $\gamma$, the optimal value appears to vary based on both the heatmap source and the problem scale. This suggests that "out-of-the-box" performance might require instance-specific tuning for maximum efficacy

---

> ### Author Rebuttal · Authors · 2026-03-30
>
> 1. **Q1: What is the practical "ant" advantage?**
>
> A heatmap scores each edge independently. Greedy decoding commits to locally attractive edges early and cannot recover when they combine into a poor global tour — this is where it fails at scale.
>
> In HeatACO, each ant builds a complete feasible tour. After each round, pheromone reinforces edges that appeared in shorter tours and weakens edges associated with detours. This gives the system instance-specific global feedback that a one-shot decoder cannot provide. The heatmap acts as a soft prior; pheromone acts as lightweight global conflict resolution. The advantage is not "human-like" reasoning, but repeated whole-tour evaluation instead of one-shot local edge selection.
>
> To verify this, we tested two simpler alternatives without pheromone feedback under the same budget (`32 × 5000` tour evaluations):
>
> - **RS**: independently samples complete tours from heatmap-filtered candidate edges, weighted by the distance term and heatmap confidence.
> - **ILS**: starts from a heatmap-guided tour, then repeatedly applies double-bridge perturbation and keeps the best tour so far; restarts from the heatmap if stuck.
>
> Both can optionally apply 2-opt (same setting as `HeatACO+2opt`).
>
> **Table 1. Mechanistic comparison on TSP10K (Gap %).**
>
> | Heatmap | RS | ILS | HeatACO | RS+2opt | ILS+2opt | HeatACO+2opt |
> |---|---:|---:|---:|---:|---:|---:|
> | AttGCN | 42.68 | 43.42 | 19.34 | 8.18 | 7.16 | 1.27 |
> | DIMES | 38.35 | 39.07 | 23.06 | 7.82 | 7.09 | 1.15 |
> | DIFUSCO | 22.37 | 23.02 | 8.82 | 4.88 | 5.00 | 1.19 |
>
> HeatACO achieves a **4–5× gap reduction** over both controls, across all heatmap predictors (also on TSP500/TSP1K; full tables in the revised appendix). Since RS and ILS use the same heatmap but lack pheromone, this confirms that pheromone feedback is a real algorithmic gain, not just an ant metaphor.
>
> In revision, we will add a pheromone-evolution visualization showing how HeatACO concentrates pheromone more quickly toward high-quality tour structure compared to vanilla MMAS.
>
> 2. **Weakness: Hyperparameter dependency.**
>
> The reviewer is right that the best `γ` depends on heatmap sharpness and local-search configuration. `γ` controls how strongly the heatmap biases ant sampling: larger `γ` sharpens the proposal toward high-confidence edges, smaller `γ` keeps it broader. When local search is enabled, over-sharpening is harmful because it reduces tour diversity and over-commits to spurious edges (e.g., on TSP10K, DIMES+2opt degrades from 1.15% at `γ=0.1` to 6.73% at `γ=1.0`).
>
> To handle this automatically, we propose a **label-free entropy-targeted rule** (Appendix G.4) that selects `γ` from the heatmap scores alone, without running decoding or requiring benchmark labels. For each node, we compute the effective support size `S(γ)` — how many candidate edges meaningfully contribute to the proposal at a given `γ`. We choose `γ` from {0.1, 0.5, 1, 2} so that the median support matches a target: `S*=8` with local search (preserving diversity), `S*=4` without (allowing stronger guidance). This is one coarse calibration per predictor/local-search configuration, reused across all scales — not tuned per instance. All other MMAS parameters stay at standard defaults.
>
> This single rule is used in **all** our reported results — TSP100/500/1K/10K with four heatmap predictors, OOD TSPLIB instances (pcb442, pr1002, pr2392), and the new clustered/Gaussian distribution-shift experiments (see our response to Reviewer cpoE, Point 1) — with competitive results in every case. In contrast, Pan et al.'s MCTS requires per-heatmap and per-scale tuning. HeatACO's tuning burden is therefore **lighter**, not heavier.
>
> 3. **Q2: CPU deployment practicality.**
>
> Once the heatmap is fixed, decoding runs on ordinary CPUs — no GPU is needed.
>
> **Table 2. CPU decoder runtimes (fixed heatmap).**
>
> | Method | TSP500 | TSP1K | TSP10K |
> |---|---|---|---|
> | MMAS+2opt | 2.49s | 4.81s | 58.58s |
> | HeatACO+2opt | 1.82–2.06s | 4.24–4.91s | 1.11–1.37m |
>
> These are multi-thread times. Even single-threaded, HeatACO+2opt (DIFUSCO) runs in 13.92s / 32.55s / 9.20m on TSP500 / TSP1K / TSP10K. For moderate-scale problems with precomputed heatmaps, deployment on ordinary hardware is practical.
>
> Without a neural heatmap, standard MMAS+2opt is even lighter (Table 2) and should already be feasible on current mobile devices. With the rapid progress of on-device SoCs and neural-network inference engines, we expect that on-device heatmap computation will become practical in the near future, making full HeatACO deployment on mobile devices increasingly viable.

---

> > ### Author Rebuttal · Reviewer_mbbF · 2026-04-03
> >
> > Thanks for comments !

---

> > > ### Author Response · Authors · 2026-04-04
> > >
> > > We sincerely thank the reviewer for confirming that the concerns have been resolved, and for the supportive evaluation throughout the review process. Your constructive comments have helped us improve the clarity and completeness of our work. We will incorporate all discussed improvements into the revised manuscript.

---

### Official Review · Reviewer_NCmh · 2026-03-13

**Soundness:** 3
**Presentation:** 3
**Significance:** 2
**Originality:** 2
**Overall Recommendation:** 4
**Confidence:** 3

**Summary:**

This paper addresses a technical bottleneck in neural combinatorial optimization (NCO), specifically for the Travelling Salesperson Problem (TSP). Some recent neural solvers work in two stages: first, a neural network produces a "heatmap," a matrix of confidence scores estimating how likely each edge (city-to-city connection) is to appear in the optimal tour; second, a decoder converts that heatmap into an actual feasible tour (a valid route with no repeated visits). This second step is the bottleneck. Greedy merge, i.e., building the tour by always picking the highest-confidence edge available, is fast but fragile at scale, because early commitments to locally attractive edges can create conflicts (e.g., visiting a city twice or leaving disconnected fragments) that cannot be repaired later. MCTS-guided local search is more powerful but computationally expensive and sensitive to how the search space is set up.

The paper proposes HEATACO, a decoder that feeds the heatmap into a Max–Min Ant System (MMAS), which is a popular metaheuristic inspired by how ants find shortest paths using pheromone trails. In particular, the heatmap is injected as a multiplicative prior (a soft bias) into the transition probabilities that govern which edge each artificial ant chooses next. The idea is to treat the heatmap as a suggestion rather than a hard filter: ants are nudged toward high-confidence edges but not locked into them, and pheromone feedback provides lightweight, instance-specific global coordination to resolve conflicts (such as degree violations or subtours) that greedy methods cannot fix. Optionally, 2-opt/3-opt post-processing (local search moves that remove crossing edges or rearrange short segments) further tightens the resulting tours.

Across four pretrained heatmap predictors (AttGCN, DIMES, UTSP, DIFUSCO) on TSP benchmarks at three scales (500, 1,000, and 10,000 nodes), HEATACO+2opt achieves optimality gaps of 0.11%/0.23%/1.15%, with CPU decoding budgets ranging from seconds to minutes. This improves over greedy decoding and reports better gaps than published MCTS decoders at similar nominal budgets, though not on matched hardware. The paper also provides a useful analysis of heatmap reliability, documenting confidence-mass concentration (most of the heatmap's probability mass sits on a small fraction of edges) and low-confidence collapse (on harder out-of-distribution instances, the heatmap loses its ability to distinguish good edges from bad ones), tested on TSPLIB instances.

**Compliance With Llm Reviewing Policy:**

Affirmed.

**Final Justification:**

I recommend weak accept, with little conviction. The authors did a good job in rebuttal: they were responsive, added new evidence, and addressed the main technical concerns from my review. In particular, the new non-ACO controls make the mechanistic story more credible, the matched-hardware MCTS comparison reduces my main fairness concern, the paired/variance analysis supports that the small 2-opt gains are not just noise, and the follow-up clarification largely resolved the ambiguity I had about the ablation setup.

These additions improved my view of the paper’s soundness, which is why I moved from weak reject to weak accept. However, my support remains quite lukewarm. Even after rebuttal, I still see the contribution as narrow and fairly incremental: a decoder-side improvement for heatmap-based TSP, with limited conceptual novelty and limited demonstrated scope beyond that setting. From my perspective, the paper does not yet add a great deal to the literature, and I would not advocate for acceptance on novelty or breadth grounds.

So my recommendation should be read in that limited sense. I think the authors improved the work meaningfully and made it technically much more convincing within its scope. If the community is interested in encouraging progress in this direction, I can support a weak accept. But if the bar is meant to emphasize broader novelty or wider significance, I would also understand a more negative decision. I also assume that the final version will incorporate the new rebuttal evidence and clarifications into the paper in an integrated way.

**Key Questions For Authors:**

1. Can you provide any controlled evidence, or a stronger argument, that pheromone feedback specifically is the key ingredient, rather than iterative stochastic search biased by a learned prior more generally? For example, even a single non-ACO comparison, such as repeated sampling from the heatmap distribution with best-tour selection, or a heatmap-biased iterated local search, would help clarify whether the gains are due to pheromone feedback itself or to guided iterative search more broadly.

2. In the Table 2 ablations, how are candidate lists constructed for the gamma=0 and beta=0 conditions? The candidate lists are built from top-ranked heatmap neighbours with distance-based fallback, so they encode both heatmap and distance information. If the gamma=0 baseline still uses heatmap-derived candidates, the heatmap is shaping the search space even with the transition-rule prior disabled; symmetrically, if the beta=0 baseline still includes distance-based fallback candidates, distance information persists even with the distance heuristic removed. Clarifying whether the candidate-list construction changes across ablation conditions would significantly affect interpretation of Table 2. If it does not change, an additional ablation varying the candidate-list construction would help cleanly separate the two channels through which heatmap and distance information enter the decoder.

3. Can you provide variance reporting and a paired instance-level analysis for the key HEATACO+LS vs. MMAS+LS comparisons? Standard deviations or confidence intervals for Tables 1 and 2 would help, but more importantly, the claim that matters is whether HEATACO+LS consistently beats MMAS+LS on the same instances. A win/loss/tie count across instances, a confidence interval over per-instance gap differences, or a form of statistical test would be persuasive. The marginal improvements with local search enabled are small enough that without instance-level evidence of consistent improvement, the difference may not be reliably meaningful.

4. How were the gamma values used in the headline tables (Tables 1 and 2) selected? Were they chosen by sweeping over the same benchmark instances used for reporting, via the label-free entropy-targeted rule, or through some held-out protocol? This is important for evaluating the method's actual tuning burden and the fairness of the "low tuning" framing relative to MCTS.

5. What happens if HEATACO is tuned with the same per-heatmap, per-scale hyperparameter effort as the MCTS decoder of Pan et al. (2025)? Conversely, how does the MCTS decoder perform with a quick default configuration on matched hardware? Answering either direction would make the comparison in Table 1 more informative and address the apples-to-apples concern.

6. Could you add a discussion of what adaptations would be needed for constrained routing problems (e.g., CVRP, Pickup and Delivery)? The "plug-and-play" framing is well-supported for cross-predictor use on TSP, but the broader "routing and logistics" language in the introduction may suggest wider applicability than what is demonstrated. A brief discussion of what would change for problems with capacity, precedence, or multi-route constraints would help readers calibrate the scope.

**Limitations:**

The authors acknowledge hardware differences in timing comparisons (Appendix B.2), which is appreciated. Other suggestions (already mentioned in my previous comments):

(1) the evaluation is restricted to Euclidean TSP, limiting generalizability claims despite "plug-and-play" framing.
I recognize that Euclidean-TSP-only results are kind of a standard in the NCO decoding literature, and I understand the practical reasons. To the paper's credit, the "plug-and-play" framing is primarily about applying the decoder across different heatmap predictors and instance distributions, which the experiments do validate well. However, the introduction also discusses "routing and logistics" applications more broadly, which may suggest to readers a wider scope than what is demonstrated. TSP has a specific structure that makes ACO decoding relatively straightforward. For problems like CVRP (capacity constraints, variable route counts, depot returns) or Pickup and Delivery (precedence constraints), it is not obvious that the MMAS transition rule with a heatmap prior transfers without significant modification. I am not asking the authors to run CVRP experiments, but I would suggest adding a discussion of what would need to change for constrained routing problems, so readers can calibrate how far the approach extends beyond Euclidean TSP.

(2) the diminishing marginal value of the heatmap prior when combined with strong local search operators is not discussed as a practical limitation;

(3) the heatmap reliability analysis proposes several diagnostics but does not validate which best predicts downstream decoding quality beyond the tested benchmarks.

**Strengths And Weaknesses:**

Strengths

Soundness: The experimental evaluation is thorough for a decoding paper. Testing across four different heatmap predictors (AttGCN, DIMES, UTSP, DIFUSCO) and three scales (500/1K/10K) with consistent protocols is valuable. The comparison includes several baseline families (greedy merge, MCTS-guided decoding) and useful ablations (Table 2), and the sensitivity analysis to gamma (Figure 4, Appendix G) provides practical guidance for practitioners.

Presentation: The writing is overall clear although there is a lot of jargon that could be better introduced for a less familiar reader. Figures are overall good and the Appendix provides solid additional details.

Significance: Section 5 (heatmap reliability analysis) provides interesting insights. The confidence-mass concentration analysis (Figure 3) and the documentation of low-confidence collapse on TSPLIB instances are useful diagnostics that go beyond the specific decoder proposed here. These observations, that optimal-tour edges cluster in a narrow mid/high-confidence band while the vast majority of candidate edges sit in a low-confidence tail, help explain why greedy decoders fail at scale and why treating the heatmap as a soft prior is more appropriate than hard thresholding.

Weaknesses

W1: Missing mechanistic isolation

The core idea is appealing: ACO sits naturally between fast-but-rigid greedy merge and powerful-but-expensive MCTS, and injecting the heatmap as a multiplicative bias into MMAS is a minimal, clean modification. I appreciate the simple design.
That said, the novelty is primarily empirical. Beyond the heatmap term H_ij^gamma in the transition rule and heatmap-derived candidate lists, everything else (pheromone update, evaporation, optional 2-opt/3-opt) is fairly standard MMAS. I want to be fair here: the paper is upfront about this, and simplicity is a virtue. But it means the experiments need to show that pheromone feedback specifically is what makes this work. Warm-starting metaheuristics with learned priors is not new. For instance, DeepACO (Ye et al., 2023) already pairs neural heatmaps with ACO, albeit with end-to-end training rather than a fixed pretrained predictor. Moving from joint training to a fixed prior is a real design choice, but it is an engineering one.
The most important missing experiment is a non-ACO baseline that would test whether pheromone feedback is actually the key ingredient, or whether any form of iterative stochastic search biased by a heatmap would do equally well. Even one alternative such as repeated sampling from the heatmap distribution keeping the best tour, or a heatmap-biased iterated local search, would be helpful. Without it, the paper compares only against greedy merge and MCTS-guided k-opt, and the question of why ACO specifically remains unanswered.

W2: Comparison fairness

The comparison with MCTS-guided decoding is not fully apples-to-apples. Table 1 is the core of the empirical evaluation, and the paper's headline claim is that HEATACO+2opt "improves over published MCTS-guided decoding under comparable or smaller budgets." I am confused with the provenance of the MCTS results in Table 1 Do they come from Pan et al. (2025), who use a 128-core CPU implementation with a fixed time budget of 0.1N seconds per instance? Did you rerun the MCTS decoder on your hardware or reproduced their results directl? The HEATACO results use 16 threads on an AMD EPYC 9634. As the authors acknowledge (Appendix B.2), "due to differences in hardware, tuning, and implementation, these numbers should be interpreted as indicative rather than strictly comparable." I appreciate this, but it undermines the quantitative comparison that the paper emphasizes in its abstract and introduction.

In addition, I could not find the exact details on how the gamma values used in the headline tables (Tables 1 and 2) were selected. The appendix presents coarse gamma sweeps and a label-free entropy-targeted selection rule, but it is unclear whether the reported results use gamma values chosen on the benchmark instances themselves, the label-free rule, or some held-out tuning protocol. This matters directly for the "practical / low tuning" framing: if gamma was selected by sweeping over the same benchmark instances used for reporting, the method's tuning burden is higher than presented. A practitioner may wonder: if I invest the same engineering effort in tuning HEATACO (per-heatmap, per-scale hyperparameters for m, I, k, alpha, beta, gamma, rho, and candidate-list construction), how much better does it get? And conversely, if I run the MCTS decoder with a quick default configuration on 16 threads, how much does it degrade? This question is very addressable.

W3: Ablation leakage

The ablations in Table 2 may not cleanly isolate what they claim to. HEATACO builds candidate lists from top-ranked heatmap neighbors with distance-based fallback, so the lists encode both heatmap and distance information. When gamma=0 "removes the heatmap" from the transition rule, heatmap-derived candidates still shape which edges ants can visit. The same applies in reverse: when beta=0 "removes distance," distance-based fallback neighbors remain. I may have misunderstood something, but Table 2 therefore does not appear to fully isolate either component because even when one is removed from the transition rule, it still influences the search through the candidate list. If the ablation baselines use different candidate-list constructions, the paper should say so; if they don't, an additional ablation varying the candidate lists would clarify interpretation. This matters because the strongest evidence that the heatmap carries useful information (e.g., 3.53% vs. 6.13% on TSP500, 8.82% vs. 38.36% on TSP10K) comes from Table 2, and its interpretation hinges on how clean the gamma=0 and beta=0 conditions actually are.

W4: Diminishing returns and missing variance

Once local search is added, the gains from the heatmap prior shrink dramatically. MMAS+2opt reaches 0.16%/0.42%/1.20% on TSP500/1K/10K; HEATACO+2opt gets 0.11%/0.23%/1.15% (-->fraction of a percentage point). With 3-opt (Table 3), MMAS+3opt is already extremely strong, and HEATACO+3opt adds only tiny changes around that level. The paper acknowledges this narrowing (Section 6.1, last paragraph), but I think it deserves more engagement: a practitioner with a good local search implementation needs to know whether a 0.05pp improvement justifies the added complexity of sourcing a neural heatmap.

This concern is compounded by the absence of variance reporting. Table 1 averages over 10 seeds but reports no standard deviations or confidence intervals. For comparisons where the heatmap clearly dominates (e.g., vs. greedy merge on TSP10K), it isn't as much of an issue. But for the comparisons that carry the paper's main claims, like 0.11% vs. 0.16% on TSP500, without paired instance-level analysis, it is hard to judge whether this difference is robust or within noise. I would really like to see something along the lines of confidence intervals for Tables 1 and 2, and provide a paired analysis (e.g., win/loss/tie across instances, or a statistical test) for the key HEATACO+LS vs. MMAS+LS comparisons.

To be clear, I am not dismissing the contribution. The heatmap prior clearly helps without local search, and the time-to-quality curves (Figure 2) seem to show real improvements. But the paper emphasizes its best-case numbers without adequately contextualizing how close plain MMAS+LS gets on its own. I would like to see this trade-off discussed more directly.

W5: Partially validated diagnostics

The heatmap reliability analysis is useful but only partially validated as a practical diagnostic framework. Section 5 documents two phenomena: confidence-mass concentration and low-confidence collapse, and relates them to decoder performance. This is valuable work, and the paper does make some prescriptive moves: it suggests monitoring Edges/N and confidence-mass profiles, proposes coarse gamma sweeps, and provides a label-free entropy-targeted gamma selection rule. However, the analysis does not establish which of these diagnostics best predicts downstream decoding quality. The CE/WCE diagnostics (Table 4, Appendix C.2) are presented as auxiliary proxies for heatmap quality, but Figure 5 suggests the relationship between CE/WCE and decoding gap is only loose, especially once decoder dynamics and local search enter. The paper acknowledges this ("CE/WCE are only auxiliary indicators"), but if they are only weak auxiliary indicators, are the proposed alternatives (Edges/N, confidence-mass profiles) better? The paper does not validate this. I found the analysis useful, but the gap between "here are some diagnostics" and "here is a validated framework for predicting decoder-ready heatmap quality" remains open.

---

> ### Author Rebuttal · Authors · 2026-03-30
>
> 1. **W1/Q1: Mechanistic isolation.**
>
> To isolate the role of pheromone feedback, we add two non-ACO controls under the same budget (`160,000` tour evaluations):
>
> - **Random Sampling (RS)**: samples complete tours from heatmap-filtered candidate edges, weighted by the distance term and heatmap confidence.
> - **Iterated Local Search (ILS)**: starts from a heatmap-guided tour, repeatedly applies heatmap-guided double-bridge perturbation and keeps the best; restarts if stuck.
>
> Both optionally apply 2-opt in each step/sampling (same setting as `HeatACO+2opt`).
>
> **Table 1. Mechanistic isolation on TSP10K (Gap %).**
>
> | Heatmap | RS | ILS | HeatACO | RS+2opt | ILS+2opt | HeatACO+2opt |
> |---|---:|---:|---:|---:|---:|---:|
> | DIMES | 38.35 | 39.07 | 23.06 | 7.82 | 7.09 | 1.15 |
> | DIFUSCO | 22.37 | 23.02 | 8.82 | 4.88 | 5.00 | 1.19 |
>
> HeatACO achieves a **4–5× gap reduction** over both controls. Since RS and ILS have access to the same heatmap but lack pheromone feedback, this directly shows that **pheromone is the key extra mechanism**.
>
>
> 2. **W2/Q4/Q5: Comparison fairness and `γ` selection.**
>
> The MCTS rows in the main paper are Pan et al.'s tuned 128-core results. We re-ran MCTS with default configuration on the same hardware as HeatACO (single-threaded AMD EPYC 9634).
>
> **Table 2. Matched-hardware single-thread comparison.**
>
> | Method (Heatmap) | TSP500 Gap % / Time | TSP1K Gap % / Time | TSP10K Gap % / Time |
> |---|---|---|---|
> | Concorde | 20.68s | 3.19m | >100h |
> | MCTS (DIFUSCO) | 0.50 / 50s | 1.17 / 1.67m | 2.19 / 16.67m |
> | HeatACO+2opt (DIFUSCO) | 0.11 / 13.92s | 0.23 / 32.55s | 0.54 / 9.20m |
>
> On same hardware, HeatACO+2opt achieves **3–5× lower gap** while running **2–4× faster**, confirming the advantage is not caused by hardware differences (W2/Q5).
>
> All `γ` values are set by a label-free entropy-targeted rule on all test instances including TSPLIB — **not by sweeping over benchmark labels** (Q4). Other MMAS parameters use standard defaults. Pan et al.'s MCTS involves per-heatmap/per-scale tuning, so HeatACO's tuning burden is **lighter**.
>
> Due to space limit, more details are in our response to Reviewer mHAB (Point 2).
>
> 3. **W3/Q2: Ablation leakage.**
>
> Candidate lists are built from all heatmap neighbours, filled with k-nearest neighbours by distance (Appendix B.1).
>
> When `γ=0`, the heatmap is not used at all and the algorithm reduces to standard MMAS. No heatmap leakage occurs.
>
> When `β=0`, the distance heuristic is removed from the transition rule but heatmap-derived candidates are kept. Distance only re-enters as a feasibility fallback when all candidates are already visited. This fallback is rare with good heatmap coverage and becomes rarer as pheromone accumulates.
>
> Thus `γ=0` fully removes heatmap influence, and `β=0` removes distance influence while preserving heatmap-based candidate selection.
>
> 4. **W4/Q3/L2: Variance and diminishing returns.**
>
> We provide variance and paired instance-level analysis (win/loss/tie, Wilcoxon signed-rank test; (+) means significantly better at p<0.05):
>
> **Table 3. Paired evidence.**
>
> | Variant | TSP500 | TSP1K | TSP10K |
> |---|---|---|---|
> | MMAS | 6.13 ± 0.07 | 7.72 ± 0.06 | 38.36 ± 0.29 |
> | HeatACO (DIMES) | 3.98 ± 0.05, 128/0/0 (+) | 5.20 ± 0.03, 128/0/0 (+) | 23.06 ± 0.12, 16/0/0 (+) |
> | MMAS+2opt | 0.16 ± 0.01 | 0.42 ± 0.01 | 1.20 ± 0.03 |
> | HeatACO+2opt (DIMES) | 0.14 ± 0.008; 73/0/55 (+) | 0.39 ± 0.01; 88/0/40 (+) | 1.15 ± 0.02; 14/0/2 (+) |
> | MMAS+3opt | 0.003 ± 0.001 | 0.06 ± 0.003 | 0.41 ± 0.008 |
> | HeatACO+3opt (DIMES) | 0.004 ± 0.001; 41/51/36 (=) | 0.06 ± 0.003; 61/0/67 (=) | 0.40 ± 0.02; 11/0/5 (+) |
>
> Without local search, HeatACO wins on every instance. With 2-opt, HeatACO+2opt is significantly better in 5 of 6 settings — **the gains are real, not noise**. Meanwhile, 2-opt barely increases runtime while 3-opt nearly triples it, so the 2-opt regime is where heatmap guidance has the most practical value.
>
> Even under 3-opt, HeatACO+3opt wins on a notable share of instances (e.g., 11/0/5 on TSP10K), showing that when the heatmap is accurate enough, it can beat MMAS+3opt. The bottleneck is current heatmap quality, not the method. Our convergence analysis confirms this: better heatmaps lead to faster convergence. This shows a great potential of promising performance upon improved heatmaps in future work.
>
>
> 5. **W5/L3: Diagnostics scope.**
>
> Section 5 is an observational analysis  meant to understand when heatmap-guided decoding works or fails — we do not claim CE/WCE, Edges/N, or confidence-mass profiles are validated predictors of downstream gap. We will state this explicitly in the revision.
>
> 6. **Q6/L1: Broader applicability.**
>
> HeatACO is applicable to ATSP and CVRP in principle, provided that (1) an accurate heatmap is available and (2) a problem-specific ACO handles the constraints. We tested only on TSP since it is currently the only problem with available heatmaps.
>
> Due to space limit, more details are in our response to Reviewer mHAB (Point 4).

---

> > ### Author Rebuttal · Reviewer_NCmh · 2026-04-02
> >
> > I thank the authors for the rebuttal. I would raise my score from weak reject to weak accept with additional clarifications with the questions below. The rebuttal already strengthened the paper on two issues I viewed as decision-relevant: mechanistic isolation and statistical support for the small local-search gains. One methodological issue around the interpretation of the ablations still appears unresolved, and I explain that below together with what I would expect to see incorporated into the final revision.
> >
> > What the rebuttal addressed well
> >
> > The mechanistic-isolation experiment (W1/Q1) was an important missing piece, and the new RS and ILS controls go a long way toward addressing it. Under a matched budget and with access to the same heatmap, these controls remain worse than HeatACO, which provides more evidence that pheromone-based ACO contributes beyond simple non-ACO iterative search.
> >
> > The matched-hardware comparison (W2/Q5) also reduces my fairness concern. Re-running MCTS on the same hardware and showing that HeatACO+2opt still achieves substantially lower gap at lower runtime is useful evidence that the main effect is not just a hardware artifact. The clarification that $\gamma$ is selected via a label-free entropy-based rule, rather than by sweeping on benchmark labels, also addresses my concern about the practical tuning burden. I still view this as weaker than a fully matched-effort tuning comparison, but it is enough to alleviate my main concern on this point.
> >
> > The paired instance-level analysis (W4/Q3) is also helpful. The new variance reporting and win/loss/tie analysis show that the 2-opt gains are mostly real rather than statistical noise, which is what I most wanted to see. The rebuttal also clarifies the practical interpretation: the gains narrow further under 3-opt, but 2-opt appears to be the more relevant operating regime.
> >
> > What remains open
> >
> > The ablation-leakage response (W3/Q2) is the one point I still find unsatisfying. The authors state that candidate lists are built from heatmap neighbors and then filled with k-nearest neighbors by distance, but then assert that when $\gamma = 0$, the heatmap is not used at all, and no heatmap leakage occurs. Can you clarify how these statements fit together cleanly? If the candidate list is still constructed from heatmap-ranked neighbors, then the heatmap continues to shape which edges ants can visit, even when the transition-rule prior is disabled. Either the candidate-list construction changes when $\gamma = 0$, in which case the paper should state this explicitly, or it does not, in which case the “no leakage” claim needs to be revisited. Please clarify this unambiguously in the revision.
> >
> > What I would expect in the revised version
> >
> > The rebuttal introduces important new evidence, and I would like to see it incorporated into the revised paper in a thorough manner. In particular:
> >
> > 1. Mechanistic isolation (RS and ILS controls). This is the strongest new evidence in the rebuttal and should be incorporated into the revision. I believe it should be shown more broadly across scales and/or predictors, or accompanied by a brief explanation of why the reported settings are representative.
> >
> > 2. Matched-hardware comparison. The single-hardware MCTS comparison should appear in the main paper or a prominent appendix table, since it directly informs one of the paper’s headline comparisons.
> >
> > 3. Variance and paired analysis. Standard deviations and win/loss/tie or equivalent paired instance-level evidence should be reported for the key comparisons in Tables 1 and 2, especially where the margins are small.
> >
> > 4. Ablation clarification. The candidate-list construction for the $\gamma=0$ and $\beta=0$ conditions should be stated unambiguously. If the candidate-list mechanism does not change across ablations, the implications for interpretation should be discussed clearly.
> >
> > 5. Distribution-shift results. The clustered and Gaussian experiments presented in rebuttal to Reviewer cpoE are valuable and would strengthen the paper if included in the revision, ideally with the same variance reporting.
> >
> > 6. Diagnostics scope (W5). The paper should explicitly state that CE/WCE and related diagnostics are observational tools rather than validated predictors of downstream decoding quality.
> >
> > 7. TSP-only scope language. The broader “routing and logistics” framing should be calibrated more carefully to maintain the paper's demonstrated scope.

---

> > > ### Author Response · Authors · 2026-04-03
> > >
> > > We sincerely thank the reviewer for the thoughtful and constructive feedback, and for recognizing the new evidence. We are glad these additions helped address the main concerns.
> > >
> > > ## Ablation clarification (W3/Q2)
> > >
> > > Thank you for pointing out the ambiguity in our rebuttal — the reviewer's reading is entirely reasonable, and we appreciate the opportunity to clarify.
> > >
> > > When `γ=0`, the candidate list is **NOT constructed by the heatmap**: it switches to pure distance-based k-nearest neighbours, which is the standard MMAS candidate list. No heatmap information is used in any part of the algorithm. The `γ=0` condition is therefore a true standard MMAS baseline with no heatmap leakage.
> > >
> > > The confusion arose because our rebuttal first described HeatACO's heatmap-based candidate list, then stated that `γ=0` has no leakage, without explicitly noting that `γ=0` uses a different candidate list. We apologize for this ambiguity.
> > >
> > > In the revision, we will:
> > > - Rename `HeatACO(γ=0)` to `MMAS` in the ablation table to remove any ambiguity.
> > > - Explicitly state that when `γ=0`, candidate lists are built from distance-based k-nearest neighbours only, matching standard MMAS.
> > >
> > > ## Planned revisions
> > >
> > > We address each of the reviewer's suggested revisions below:
> > >
> > > 1. **RS/ILS controls**: Already completed across all scales (TSP500/1K/10K) and all four heatmaps. Only TSP10K with DIMES/DIFUSCO was shown in the rebuttal due to the character limit; the full results are consistent — HeatACO outperforms both controls by a wide margin in every case. Full table attached below.
> > >
> > > 2. **Matched-hardware comparison**: Will be added to the main paper.
> > >
> > > 3. **Variance and paired analysis**: Standard deviations and win/loss/tie will be reported for key comparisons in Tables 1 and 2.
> > >
> > > 4. **Ablation clarification**: As described above.
> > >
> > > 5. **Distribution-shift results**: Clustered and Gaussian experiments (from our response to Reviewer cpoE) will be included with variance. All HeatACO vs MMAS comparisons are statistically significant (Wilcoxon signed-rank, p<0.05). Full table attached below.
> > >
> > > 6. **Diagnostics scope (W5)**: Will explicitly state that CE/WCE and related diagnostics are observational, not validated predictors of downstream quality.
> > >
> > > 7. **TSP-only scope language**: Broader framing will be calibrated to match the paper's demonstrated scope. **We have also completed new ATSP experiments** (ATSP50/100 using UniCO's MatDIFFNet heatmap), where HeatACO+2opt achieves 0.26%/0.63% gap with the same parameters and γ selection rule as TSP. We kindly refer the reviewer to our response to Reviewer mHAB for details.
> > >
> > > ---
> > >
> > > We sincerely appreciate the reviewer's careful and constructive engagement. We are confident that the clarifications and additional evidence — together with the planned revisions — fully address the remaining concerns.
> > >
> > >
> > > ---
> > >
> > > ## Appendix
> > >
> > > ## A1. Full RS/ILS results
> > >
> > > All methods use the same budget (160,000 tour evaluations).
> > > Standard deviation is omitted here due to space limit, but will be inlucded in the revision.
> > >
> > > **Table A1a. TSP500 (Gap %).**
> > >
> > > | Heatmap | RS | ILS | HeatACO | RS+2opt | ILS+2opt | HeatACO+2opt |
> > > |---|---:|---:|---:|---:|---:|---:|
> > > | AttGCN | 54.74 | 57.80 | 3.53 | 4.76 | 2.79 | 0.17 |
> > > | DIMES | 38.76 | 41.55 | 3.98 | 4.37 | 2.76 | 0.14 |
> > > | UTSP | 113.08 | 116.81 | 4.39 | 4.76 | 2.79 | 0.16 |
> > > | DIFUSCO | 21.94 | 25.22 | 0.80 | 1.83 | 2.08 | 0.11 |
> > >
> > > **Table A1b. TSP1K (Gap %).**
> > >
> > > | Heatmap | RS | ILS | HeatACO | RS+2opt | ILS+2opt | HeatACO+2opt |
> > > |---|---:|---:|---:|---:|---:|---:|
> > > | AttGCN | 31.60 | 33.50 | 4.97 | 5.57 | 4.33 | 0.44 |
> > > | DIMES | 30.46 | 32.39 | 5.20 | 5.69 | 4.33 | 0.39 |
> > > | UTSP | 59.79 | 62.08 | 6.42 | 6.42 | 4.43 | 0.42 |
> > > | DIFUSCO | 15.43 | 17.33 | 1.20 | 2.41 | 2.79 | 0.23 |
> > >
> > > **Table A1c. TSP10K (Gap %).**
> > >
> > > | Heatmap | RS | ILS | HeatACO | RS+2opt | ILS+2opt | HeatACO+2opt |
> > > |---|---:|---:|---:|---:|---:|---:|
> > > | AttGCN | 42.68 | 43.42 | 19.34 | 8.18 | 7.16 | 1.27 |
> > > | DIMES | 38.35 | 39.07 | 23.06 | 7.82 | 7.09 | 1.15 |
> > > | DIFUSCO | 22.37 | 23.02 | 8.82 | 4.88 | 5.00 | 1.19 |
> > >
> > > ## A2. OOD results with std
> > >
> > > All HeatACO vs MMAS comparisons below are statistically significant (Wilcoxon signed-rank, p<0.05).
> > >
> > > **Table A2a. Clustered TSP500 (Gap %).**
> > >
> > > | Heatmap | HeatACO | HeatACO+2opt |
> > > |---|---:|---:|
> > > | MMAS (no heatmap) | 5.42 ± 1.41 | 0.07 ± 0.10 |
> > > | AttGCN | 4.80 ± 1.42 | 0.09 ± 0.10 |
> > > | DIMES | 6.56 ± 2.12 | 0.18 ± 0.20 |
> > > | UTSP | 7.49 ± 3.04 | 0.19 ± 0.25 |
> > > | DIFUSCO | 4.46 ± 1.74 | 0.11 ± 0.15 |
> > >
> > > **Table A2b. Gaussian TSP500 (Gap %).**
> > >
> > > | Heatmap | HeatACO | HeatACO+2opt |
> > > |---|---:|---:|
> > > | MMAS (no heatmap) | 6.46 ± 1.23 | 0.22 ± 0.14 |
> > > | AttGCN | 4.58 ± 0.72 | 0.19 ± 0.11 |
> > > | DIMES | 4.71 ± 1.10 | 0.17 ± 0.11 |
> > > | UTSP | 19.29 ± 2.83 | 2.86 ± 0.60 |
> > > | DIFUSCO | 5.06 ± 1.03 | 0.20 ± 0.12 |

---

### Decision · Program_Chairs · 2026-04-30

**Decision:**

Reject

**Comment:**

The reviewers were mostly on the border about this paper. There are several key weaknesses of this work that led the discussion towards the negative side:

1. The paper does not offer a significant contribution. The paper is basically a heatmap-based NCO approach that harnesses a specialized ACO algorithm for its search over the heatmap recommendations. I note that this alone is not enough, at least in my opinion, to reject the paper, because even small contributions can be important. However, it weighs negatively on the work.

2. Experimentally, the work only addresses the TSP. In the rebuttal, the authors argue the approach is applicable to the VRP. I believe this is likely the case. However, the question remains how this approach should work on anything else, especially problems with interesting constraints. Much of the NCO literature in routing at least looks at problems with time windows or other constraints. This is completely missing, so I do not see the generality of the approach. Indeed, it relies heavily on domain-specific operators for the search, so one wonders how much development effort must be put into making this approach work for other problems.

3. The reviewers point out quite a few experimental issues, including missing baselines. Table 1 is also problematic in terms of the times for each method being so different. Many NCO methods now are trying to compare with similar computational effort. The interpretation of that term can vary greatly, but I do not think the table is acceptable in its current form.

Put together, the authors have an interesting idea that unfortunately does not quite make the cut for ICML at this time.